# Very High-Energy Electron Therapy Toward Clinical Implementation

**DOI:** 10.3390/cancers17020181

**Published:** 2025-01-08

**Authors:** Costanza Maria Vittoria Panaino, Simona Piccinini, Maria Grazia Andreassi, Gabriele Bandini, Andrea Borghini, Marzia Borgia, Angelo Di Naro, Luca Umberto Labate, Eleonora Maggiulli, Maurizio Giovanni Agostino Portaluri, Leonida Antonio Gizzi

**Affiliations:** 1Intense Laser Irradiation Laboratory, National Institute of Optics, National Research Council of Italy, 56124 Pisa, Italy; simona.piccinini@ino.cnr.it (S.P.); gabriele.bandini@ino.cnr.it (G.B.); luca.labate@ino.cnr.it (L.U.L.); leonidaantonio.gizzi@ino.cnr.it (L.A.G.); 2Institute of Clinical Physiology, National Research Council of Italy, 56124 Pisa, Italy; mariagrazia.andreassi@cnr.it (M.G.A.); andrea.borghini@cnr.it (A.B.); 3ASL Brindisi, Radiotherapy, 72100 Brindisi, Italy; marzia.borgia@asl.brindisi.it; 4ASST Papa Giovanni XXIII Hospital, Radiotherapy, 24127 Bergamo, Italy; adinaro@asst-pg23.it (A.D.N.); mportaluri@asst-pg23.it (M.G.A.P.); 5National Institute for Nuclear Physics, 56127 Pisa, Italy; 6ASL Brindisi, Medical Physics, 72100 Brindisi, Italy; eleonora.maggiulli@asl.brindisi.it

**Keywords:** external beam radiotherapy, VHEE, FLASH radiotherapy

## Abstract

This paper explores the potential of Very High Energy Electron (VHEE) beams, with energies between 50 and 400 MeV, as a promising option for cancer treatment. VHEE beams combine deep tissue penetration, sharp beam edges, easy manipulation using magnetic components, and cost-effectiveness, making them highly precise and versatile for tumor targeting. They also enable FLASH radiotherapy with VHEE (FLASH-VHEET), which may reduce side effects by sparing healthy tissues while effectively treating deep tumors. However, clinical adoption requires advancements in accelerator technology, treatment planning systems, and validation of treatment protocols. FLASH-VHEET also introduces challenges related to time-dependent dose delivery. This paper reviews recent progress in VHEE research, focusing on dosimetric properties, beam delivery, radiobiological effects, and clinical applications. By addressing these aspects, this study provides a foundation for advancing VHEE therapy towards clinical implementation.

## 1. Introduction

Electron beams, with energies ranging from 5 to 20 MeV, are widely used in the radiotherapy (RT) treatment of shallow tumor volumes [1,2]. However, due to their limited penetration ability, they are not suitable for deep-seated tumors (>10 cm). Additionally, they exhibit significant lateral spread, especially at the depths of their practical range, and their air scattering is high enough to prohibit pencil beam scanning.

Around two decades ago [3,4,5], research suggested that by significantly increasing the energy of electrons, all the principal limitations of low-energy external beam RT (EBRT) could be adequately addressed. In this context, the study of very high energy electron (VHEE) beams, ranging from 50 to 400 MeV, opened the possibility of developing a new type of electron EBRT, termed VHEET, which could potentially treat deep-seated tumor sites. From an academic perspective, the following key questions arose from the initial studies: Is it worthwhile to pursue the clinical development of VHEE? Does VHEE offer significant advantages over well-established X-ray RT? A stimulating discussion on these questions can be found in the 2004 point/counterpoint debate titled *Are VHEE beams an attractive alternative to photon IMRT* (Intensity-Modulated RT)*?* by Papiez and Bortfield [6], which first highlighted the pros and cons of VHEET. The listed advantages included greater potential for sparing healthy tissue compared to photons, reduced sensitivity to anatomical inhomogeneities, and ease of electromagnetic (EM) scanning in pencil beam configuration. On the other hand, the noted disadvantages included a relatively flat dose distribution with a high entrance dose and a significant yield of secondaries (mainly neutrons). Most of these issues have been addressed in subsequent studies.

From a chronological perspective, the rising interest in VHEET has paralleled the growing attention to the so-called FLASH effect [7,8]. Characterized by ultra-high dose–rate (UHDR) irradiation of 40 Gy/s, FLASH is now considered one of the most promising breakthroughs in RT. Although the mechanism behind the FLASH effect remains unclear as of this writing, the benefits of FLASH-RT have been proved: at UHDR, normal tissue toxicity is reduced while anti-tumor efficacy is maintained. In this context, VHEE FLASH-RT could represent an exceptional treatment solution, as it would combine the normal tissue-sparing capability of FLASH with the unique dosimetric advantages of VHEET. Therefore, the use of VHEE beams for FLASH-RT is highly anticipated in the near future, with numerous studies currently focused on developing reliable clinical platforms.

Despite the highly attractive properties of VHEE beams for therapeutic use, their successful clinical implementation critically depends on academic research [9]. To this end, developing safe, stable, and compact VHEE beam delivery systems is crucial. At the actual status of research, VHEET acceleration systems primarily rely either on research-based radiofrequency (RF) linear accelerators [10,11,12] or laser-driven [13,14,15,16] plasma accelerators. However, a more systematic investigation as well as major engineering and industrial efforts are required before compact and clinically compliant prototypes will emerge from these technologies. The development of VHEET delivery systems also calls for the advancement of fast and accurate VHEET treatment planning systems [17]. Moreover, the clinical translation of VHEET still necessitates the development of accurate and practical secondary dosimeters [18], as well as a deeper understanding of VHEE radiobiology [19].

This paper reviews the most recent developments in VHEET-related research and is organized as follows: Section 2 covers the physical properties of VHEE beams. Section 3 evaluates dose contributions from secondary neutrons and induced radioactivity. Section 4 discusses FLASH-VHEET under UHDR conditions. Section 5 examines advancements in VHEE RF linac-based and laser-driven accelerators. Section 6 highlights the benefits of beam focusing, while Section 7 addresses VHEE dosimetry. Section 8 reviews recent findings in VHEE radiobiology. Section 9 presents the current status of VHEE treatment planning systems and provides an overview of preliminary clinical assessment results across various tumor sites.

## 2. Physical and Dosimetric Properties of VHEE Beams

This section evaluates the key properties of VHEE beams. The terms divergent and non-divergent/collimated beams will be used frequently throughout the paper, with definitions provided in Table 1.

### 2.1. Longitudinal Dose Distribution

The main properties of VHEE beams were first evaluated in the early 2000s by DesRosiers et al. [3], who highlighted their potential clinical advantages compared to photon RT. In general, the main longitudinal dosimetric parameters of an electron beam are as follows: (a) $d_{max}$, the depth at which the maximum dose deposition, $D_{max}$, occurs; (b) PFO (the proximal fall-off) at 90% of $D_{max}$; and (c) TR, the therapeutic range at 90% of $D_{max}$. As shown in Figure 1a, for an ideal parallel beam with no divergence (i.e., collimated) and a transverse extent greater than the electron mean free path, the VHEE depth–dose curve presents three distinct regions: (1) a build-up region (surface to PFO), where the dose increases from its surface value toward 90% of $D_{max}$; (2) a wide dose peak region (PFO to TR), where all dose values are above 90% of $D_{max}$; and (3) a rapid fall-off region (beyond the TR), where the dose quickly decreases. At depths greater than the maximum VHEE range, the tail of the depth–dose distribution is due to Bremm. photons. Ideally, the tumor should be located in the dose peak region, with both the build-up and fall-off regions maintained as steep as possible to effectively spare the skin and the healthy tissues surrounding the tumor site. Bohlen et al. [20] systematically examined the dosimetric properties of VHEE beams. For a parallel beam, they found that both the PFO and the TR increase with energy in a linear and non-linear fashion, respectively. For beams smaller than 10 × 10 ${\mathrm{cm}}^{2}$, both PFO and TR markedly increase with beam size. For beams larger than this value, the same trend is observed, but the growth with beam size is smoother. With a dose peak region potentially extending up to 40 cm (for a 250 MeV beam with a 15 × 15 ${\mathrm{cm}}^{2}$ beam size), it is evident that VHEE beams ensure solid and adequate penetration through the patient’s body. Figure 2a,b show the percentage depth dose distribution (PDD) in water (with vacuum beforehand) of ideal VHEE beams, ranging in energy from 50 to 400 MeV in steps of 50 MeV, for an entrance beam diameter of 1 and 10 cm, respectively.

Unlike low-energy electrons, the divergence of VHEE due to scattering in air is quite negligible. However, a relatively extended source-to-surface distance (SSD) may cause the beam to diverge. As observed by Bohlen et al., for a divergent (i.e., non-collimated) VHEE beam, the PFO lies almost at the surface, causing the build-up region to shift within the peak region. In this situation, the TR values show almost no energy or beam size dependencies, resulting in smaller TR values compared to those in the collimated setup, with the difference increasing as the beam energy increases.

### 2.2. Transversal Dose Distribution

The transversal profile of an electron beam can be characterized by the lateral penumbra (LP), i.e., the distance between the 90% and 20% intensity levels, and the beam width (BW), i.e., the width at 90% of the maximum dose value at a given depth. Intensity levels are typically normalized to the maximum dose value. For a collimated 200 MeV VHEE beam in water, the LP and BW values are depicted in Figure 1b–d at depths of $d_{max}$, 20, and 30 cm, respectively. Generally, the lateral penumbras of VHEE beams increase with depth. For VHEE beams with energies higher than 100 MeV, this increase is very gradual at shallow depths, as faster electrons experience less scattering. However, penumbras grow considerably as electrons propagate to greater depths. Lateral scattering, which is relatively higher further away from the central axis, causes a disparity between the *on-axis* and *integrated* PDDs, as also observed from the difference between the main plots and the inset plots in Figure 2. This disparity is more pronounced in VHEE beams compared to photons and protons; nonetheless, all significant PDD properties, shapes, and peak positions remain unaltered.

**Figure 2 cancers-17-00181-f002:**
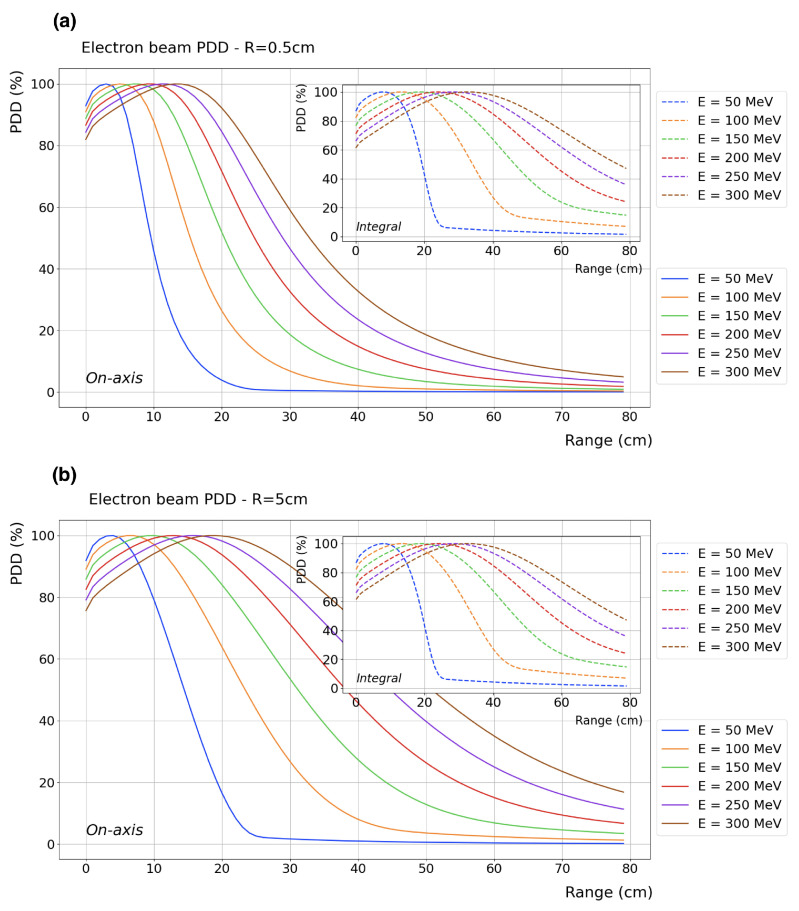
Longitudinal *on-axis* (main) and *integral* (insert) dose distributions of collimated beams impinging on a water phantom (with vacuum beforehand). The beam energy varies from 50 to 300 MeV in 50 MeV increments. The beam diameters are (**a**) 1 cm and (**b**) 10 cm.

Beam broadening at any given depth significantly decreases with increasing beam energy and only slightly decreases with decreasing beam sizes. In comparison to photon beams, VHEE beam penumbras are relatively narrower at depths of less than 5 cm and relatively wider at depths greater than 10 cm. However, since the photon PDD decreases more rapidly, the central axis dose is lower for photons than for VHEE beams at greater depths. In other words, in absolute terms, there is more spread in photon beams than in VHEE beams, which is not fully reflected by the values of relatively normalized penumbras.

For collimated beams, the BW significantly decreases with depth. Conversely, beam divergence leads to a proportional increase in the BW at the phantom entrance, thus broadening the beam. As depth increases, the BW widening from divergence is counteracted by the predominant amplitude of scattering [21]. Consequently, there is generally no beam size reduction with depth.

### 2.3. Surface Dose

For a collimated beam, the surface dose to the $D_{max}$ ratio decreases with increasing beam energy and/or enlarging beam size. This reduction is due to the greater contribution of secondary electrons and the longer build-up region of higher energy VHEE beams. However, at its best, the surface dose only lowers to about 66% of $D_{max}$ for a 250 MeV beam energy and a 10 × 10 ${\mathrm{cm}}^{2}$ beam size. Conversely, the surface dose to the $D_{max}$ ratio for a divergent VHEE beam ranges from 90 to 95%. In its current form, the surface-sparing potential of single VHEE beams is substantially lower than that of single-photon beams; indeed, the photon surface dose is equal to or even lower than 20% of $D_{max}$. To circumvent this, the implementation of focusing systems along the VHEE beamline is under evaluation, as these systems could reduce the entrance dose.

When multiple-beam arrangements are planned, the high VHEE entrance dose becomes less of a concern. In fact, the VHEE relative surface dose decreases to 60–80% with two parallel beams and to less than 50% with two perpendicular beams. Increasing the number of beams further reduces the surface dose. For example, the VHEE surface dose to the $D_{max}$ ratio decreases to less than 30% when four beams are employed. Conversely, with two parallel opposed photon beams, the surface dose is more than doubled.

### 2.4. Insensitivity to Density Variations

Due to the lack of electronic disequilibrium, VHEE beams exhibit robustness against density variations and anatomical changes, making them a promising option for treating highly inhomogeneous tumors. The dosimetric effects of inhomogeneities on VHEE beam dose profiles were studied in silico by Papiez et al. [22] and Zhou et al. [23], and by Lagzda et al. [24], both in silico and experimentally. Papiez et al. modeled a 200 MeV VHEE beam and compared it with a 15 MV photon beam in a water phantom scenario. They introduced a 2 cm spherical air cavity positioned tangentially to the beam path at a depth of 10 cm. At the edge of the cavity, a 50% decrease in dose was observed for the photon beam, whereas no significant dose variation was seen for the VHEE beam. The study by Lagzda et al. involved delivering a 156 MeV VHEE beam onto a water phantom containing slabs inserted at a depth of 2 cm. Various slab materials were chosen to simulate the dosimetric properties of soft tissue, water, and cortical bone, covering a density range from 0.001 to 2.2 g/${\mathrm{cm}}^{3}$. For each slab material, simulations compared the dose profiles of VHEE beams with those of therapeutic photons and protons under the same conditions of material heterogeneity. The results indicated that, for all embedded materials, the on-axis dose variation of VHEE beams was below 5–8%. In comparison, photon and proton dose variations were found to be as high as 74 and 100%, respectively. Lastly, using the GATE Monte Carlo (MC) toolkit, Zhou et al. evaluated the influence of inhomogeneities on 200 MeV VHEE beams—both focused and collimated—6 MV CyberKnife beams, and 150 MeV proton beams. A spherical air or rib-bone insert was embedded within a water phantom before the dose peak, along or slightly below the beam axis. Compared to photon and proton beams, VHEE beams, especially the focused ones, showed almost no dosimetric impact. Conversely, extended dose under- and over-shoot were observed for photon and proton beams when rib and air inserts were respectively placed in the two positions.

### 2.5. Electromagnetic Scanning

VHEE can be easily manipulated by magnetic components. The charge-to-mass ratio of electrons is 1836 times higher than that of protons [25], resulting in a magnetic rigidity that is lower by a factor of approximately 3–4 compared to protons of similar energy [9,26]. Relative to protons, this means that electrons can be steered and accelerated with a substantially lower intensity magnetic field, leading to a smaller technological footprint. By applying EM scanning, VHEE beams could be precisely steered along both the transverse and perpendicular planes, facilitating advanced control over beam delivery, including intensity-modulated VHEE therapy (IM-VHEET).

Because of their magnetic rigidity, electrons are highly susceptible to the Lorentz force in a magnetic resonance (MR) field, making MR-guided VHEE RT particularly challenging. Nevertheless, Yang et al. [27] proposed a potential solution, MR-PVHEE, which comprises a steering device and an MR imaging device. Using TOPAS simulations, the authors demonstrated that the steering device, i.e., a triple-stage electromagnetic steering coil set, can successfully prevent VHEE beam trajectory twisting and deflection due to the MR field.

## 3. Radiation Protection in VHEE Radiotherapy

### 3.1. Secondary Radiation Produced by the VHEE Beam in the Patient

#### 3.1.1. Neutrons

From a radiobiological perspective, neutron emission is of particular concern due to its high linear energy transfer (LET) and relative biological effectiveness (RBE), which lead to significant radiation damage. In the VHEE energy region, neutrons are primarily produced through nuclear fission initiated by Bremsstrahlung (Bremm.) photons. The Bremm. photon fluence decreases monotonically with energy, forming a broad spectrum that extends to the maximum value, the Bremm. endpoint. The Bremm. photon’s angular distribution is sharp, with the main emission occurring along the beam direction. Regarding neutron production and considering the Bremm. spectrum, two key energy regions can be identified. The first is the region of giant dipole resonance (GDR) (E < 50 MeV). The second, more relevant for high-Z materials, is the region of photon energies higher than that of the GDR:The giant dipole resonance (GDR) [28] occurs at photon energies equal to the (γ,n) reaction threshold: 10 to 19 MeV for low-Z nuclei (up to Z = 20, such as H, C, N, and O), and 4 to 6 MeV for high-Z nuclei (above Z = 20, such as Ca). During this process, the photon interacts with the nucleus as a whole, followed by the evaporation of neutrons from the excited nucleus.Above the GDR energy range, photons interact with nuclear clusters, initiating a nuclei–nuclei shower within the nucleus before neutron evaporation occurs. During this shower, parts of the nucleus gain energies that exceed the nuclear potential and are subsequently expelled.

The main channels of neutron production are the (γ,n), (γ,p), (γ,2n), and (γ,pn) reactions, with the (γ,2n) and (γ,pn) reactions being one order of magnitude less frequent than the (γ,n) reaction. The GDR can be assumed to yield neutrons exclusively. The angular distribution of GDR-produced neutrons is isotropic, with a Maxwellian energy distribution. In materials of interest for RT, considering both photon energy regions, the overall neutron distribution is nearly isotropic. The neutron yield increases as the upper limit of the Bremm. spectrum increases. The neutron spectrum typically averages a few MeV, peaks in the low-energy region, and rapidly decreases with increasing energy. The neutron quality factor for RBE estimation is conservatively set at 10 [29].

Using the Swanson theory of (γ,n) cascade [30], DesRosiers et al. [3] estimated the neutron fluence to be 0.027 and 0.037 neutrons per incident electron (n/e) for VHEE beams of 150 and 200 MeV, respectively. These values are several orders of magnitude higher than those reported in recent studies. Using FLUKA MC code, Subiel et al. [31] reported neutron yields on the order of $10^{-5}$ n/e for a 165 MeV VHEE beam, while Kokurewicz et al. [32] predicted yields of $10^{-4}$ n/e for a 200 MeV VHEE beam. Considering an SSD of 100 and 5 cm, using TOPAS MC simulations, Masilela et al. [33] estimated an n/e yield of 7 × $10^{-8}$ and 2 × $10^{-7}$ for a 200 MeV beam at depths of 0 and 30 cm within a water phantom. These authors further evaluated the differences in n/e values based on the various Geant4/TOPAS physics lists employed in the simulations, finding good agreement between the BERT, BIC, and INCLXX models.

According to DesRosiers et al., a VHEE beam with 2 × $10^{9}$ electrons (2 Gy at $d_{max}$) results in a neutron flux of 5.5 × $10^{7}$ (150 MeV) and 7.3 × $10^{7}$ (200 MeV) per ${\mathrm{cm}}^{2}$, leading to a 0.2% dose increase due to neutron fluence at $d_{max}$. Conversely, Kokurewicz et al. [32] reported about $10^{5}$ neutrons per 2 Gy fraction at the target. By means of comparison, Howell et al. [34] estimated the neutron fluence due to the secondary neutron emission for a low-energy clinical electron beam (Varian 21EX linac, 20 MV nominal energy, 10 × 10 ${\mathrm{cm}}^{2}$ beam). A fluence of 1.69 × $10^{5}$ neutrons (${\mathrm{cm}}^{2}$
${\mathrm{MU}}^{-1}$) was reported at $d_{max}$. In general, all findings agreed that neutron contamination has a negligible effect on dose deposition.

#### 3.1.2. Radioactive Isotopes

Induced radioactivity is also caused by reactions initiated by Bremm. photons. Based on Cember’s theory [35], DesRosiers et al. [3] theoretically estimated a 0.01% dose increment from induced O, C, and N radioactivity when using a 150 MeV VHEE beam. Using the FLUKA MC code, Subiel et al. [31] reported an averaged equivalent dose–rate of 96 and 3 pSv $s^{-1}$ at the front wall of a water phantom, and 458 and 38 pSv $s^{-1}$ at the rear phantom wall, at 1 and 20 min post-irradiation time, respectively. The 165 MeV VHEE beam activity was attributed to C11 ($t_{1/2}$ = 22.3 min) and O15 ($t_{1/2}$ = 122.4 s) at one minute, and C11 only at 20 min. Due to the higher Bremm. production, dose–rate values increased with depth. Kokurewicz et al. [32] assessed activation from VHEE interaction with dense body materials, such as bone and skeletal muscle, identifying isotopes like C10, Be11, N14, and Ne23. The equivalent dose, which represents the stochastic health effects of low-level ionizing radiation (e.g., radiation-induced cancer and/or genetic damage), due to induced radioactivity was found to be insignificant, and, at 10 minutes post-irradiation, it was even lower than the natural background (∼100 pSv $s^{-1}$).

### 3.2. Ambient Dose in VHEET Treatment Rooms

Masilela et al. [33] investigated the ambient dose outside a water phantom within a modeled treatment room surrounded by semi-infinite concrete walls. The neutron yield in ambient air decreased with distance from the water phantom but increased in proximity to the walls due to interactions occurring there. The highest ambient dose was estimated to be ∼0.17 mSv/Gy, which is comparable to that of proton therapy treatments [36]. In conclusion, no additional shielding is envisaged for future VHEET treatment rooms.

## 4. VHEE FLASH Radiotherapy

At its core, FLASH-RT involves the delivery of the radiation dose at ultra-high dose–rates (UHDRs) [8,37,38,39,40,41]. Generally, the FLASH-RT mean dose–rate is ≥40 Gy/s, with an optimal rate of around 100 Gy/s [42], whereas the conventional (CONV) dose–rate is ≥0.01 Gy/s. However, the full FLASH effect is more complex and depends on other interdependent temporal parameters [43]. Ideally, to elicit the FLASH effect, (1) the beam should be pulsed at a frequency of about 100 Hz, (2) the dose-per-pulse should be ≥1 Gy, and (3) the dose–rate within the pulse should be ≥ than $10^{6}$ Gy/s [44]. Together, these parameters should provide a total FLASH-RT delivery time ranging from microseconds to hundreds of milliseconds; a CONV-RT treatment typically requires several minutes [45]. To clarify, Figure 3 depicts the inter-dependent temporal parameters characterizing an entire RT treatment, a single RT fraction, and a single pulse, for both the CONV and FLASH-RT modalities. Compared to CONV modalities, the rationale for FLASH-RT lies in its ability to offer relative protection to normal tissue while preserving the anti-tumor efficacy of CONV-RT. Clinically, leveraging the FLASH effect is promising for radio-resistant tumors or tumor sites where coverage with CONV-RT is effective but causes significant normal tissue damage [44].

A major challenge in translating FLASH-RT to the clinic involves developing optimal technological systems capable of delivering FLASH-RT. Low-energy electrons were the first type of radiation whose feasibility at UHDR was assessed under pre-clinical and clinical conditions. The Kinetron (CGR MeV, Paris, France) [46,47] and Oriatron eRT6 [45,48] (PMB, Peynier, France) experimental electron linacs, developed at Orsay and Lausanne University Hospital, respectively, are the most notable FLASH delivery systems. These systems offer FLASH irradiation with energy ranges of 4–5 and 4.9–6 MeV, respectively, featuring millisecond macro-pulses and an intra-pulse dose–rate of $10^{6}$ Gy/s. In 2019, the Oriatron eRT6 was used to perform the first FLASH-RT treatment on a patient with cutaneous lymphoma [49].

Several efforts have been made to tune existing clinical linacs for low-energy electron FLASH-RT experiments. A scheme to achieve a mean dose–rate between 30 and 300 Gy/s, and specifically 74 Gy/s, was demonstrated on an Elekta Precise linac [50], as well as Varian Clinac 21EX [51] and 23EX [52] linacs, respectively. Compared to the experimental ones, clinical linacs offer beam energies, 8 to 20 MeV in range. However, due to their geometrical and dosimetric properties, they are suitable only for small animal experiments.

The procedure for transforming an IORT linac into a FLASH machine was also evaluated, with investigations performed by the same group on the NOVAC C7 (3, 5, 7, 9 MeV) [53] and Mobetron (6 MeV) [54] systems. Based on this experience, in collaboration with the SIT company [55], the ElectronFlash was designed—a compact S-band standing wave linac operating in FLASH modality at 5 and 7 MeV [56,57].

Research accelerators, such as the electron linear accelerator for beams of high brilliance and low emittance (ELBE) at HZDR in Dresden, Germany [58,59], and the Advanced Rare Isotope Laboratory (ARIEL) at TRIUMF in Vancouver, Canada [60], deliver pulsed electron beams with energies of up to 40 MeV and 30 MeV, respectively, and may provide access to ultra-high pulse dose–rates suitable for FLASH-RT.

VHEE FLASH-RT represents a unique treatment solution, as it combines the normal tissue-sparing capability of FLASH with the unique dosimetric advantages of VHEET. From a technological standpoint, the development of FLASH-RT VHEE systems, like the development of RT VHEE systems at CONV dose–rates, is currently limited to a few research infrastructures [61]. Recently, radiobiological assessments of VHEE beams at UHDR have been conducted [62,63], while novel dosimeter concepts capable of withstanding UHDR were introduced [18,64]. These studies lay the groundwork for further investigations into VHEE FLASH-RT.

## 5. Accelerators for VHEE Beams

VHEE delivery systems fall under two main categories: (1) RF linear accelerators and (2) laser–plasma accelerators. Several RF linac-based facilities currently produce VHEE beams, supporting both CONV and FLASH studies. However, these large academic setups lack delivery control features like rotating gantries and EM steering [43]. To address this, more compact RF linac platforms for FLASH-RT are also being developed. Laser–plasma accelerators, with their high-gradient energy boosts, offer compact alternatives for VHEE generation. Their short bunch durations and high charge per pulse enable high peak dose rates, making them ideal for UHDR investigations. Figure 4 shows the global distribution of RF linac-based (green), laser-based (red), and RF linac-based compact FLASH-dedicated (yellow) VHEE accelerators. For a broader overview of laser systems, including those not involved in VHEET research, the reader may refer to [65].

### 5.1. RF Linac-Based VHEE Beams Accelerators

Figure 5 schematically depicts the bare-bone structure of an RF linac. The particle source, typically powered by a high-voltage supply, generates charged particles and injects them into the beamline. A common electron source is a photon–electron gun. To prevent collisions between the accelerated electrons and air molecules, the source and all other beamline components are housed within a vacuum chamber. The length of the vacuum pipe containing the beamline can range from meters to thousands of meters, depending on the application. The electrons to be accelerated pass through a series of hollow cylindrical electrodes, which are progressively spaced farther apart as they move away from the source. This design ensures that the electrons pass through each electrode in half a cycle of the accelerating voltage. The open-ended cylindrical electrodes are powered by a microwave source, such as a klystron or magnetron, which generates an RF AC voltage on the order of thousands of volts. This setup applies opposite voltages to successive electrodes. The RF source can operate in various frequency bands, including L-band (1–2 GHz), S-band (2–4 GHz), C-band (4–8 GHz), and X-band (8–12 GHz). Low-power accelerators typically use a single RF source, while high-power machines comprise separate amplifiers, all synchronized by a frequency modulator. After passing through the final electrode, the electrons reach the beam’s diagnostic and manipulation area.

#### RF Linac-Based VHEE Beam Accelerators: Facilities Overview

The RF linac-based facilities, capable of producing stable, 60 to 220 MeV quasi-monoenergetic VHEE beams, include the following:CERN Linear Electron Accelerator for Research (CLEAR), CERN, Switzerland [80,81,82].Next Linear Collider Test Accelerator (NLCTA), SLAC, Stanford, USA [83,84,85,86].Photo Injector Test Facility at DESY in Zeuthen (PITZ), DESY, Germany [87,88,89].Accelerator Research Experiment at DESY in SINBAD (ARES), DESY, Germany [90,91,92].Sources for Plasma Accelerators and Radiation Compton with Laser And Beam (SPARC), INFN-LNF, Italy [93,94].Platform for Research and Applications with Electrons (PRAE), Paris-Sud University, France [95,96].

An overview of beam parameters, where available, is provided in Table 2.

In addition to the facilities mentioned above, two novel and compact RF linac platforms have been specifically designed for FLASH-RT: the compact C-band system (Sapienza University/INFN-LNF, Italy) [10] and the pluri-directional high-energy agile scanning electronic radiotherapy (PHASER) platform (SLAC, USA) [11]. The compact C-band system can produce VHEE beams with an energy value of 130 MeV, with a planned upgrade to reach 160 MeV. MC FLUKA simulations have demonstrated delivered pulse dose rates exceeding $10^{6}$ Gy/s and doses per pulse of up to 200 Gy at buildup. Once commissioned and fully operational, the system will support preclinical FLASH-VHEET investigations. The PHASER system, on the other hand, is being developed primarily for FLASH IMRT. However, a planned PHASER upgrade aims to enable the production of 100–200 MeV VHEE beams at UHDRs.

### 5.2. Laser-Driven VHEE Beams Accelerators

As depicted in Figure 6, the initial physical process of a laser–plasma VHEE accelerator involves focusing an ultra-short, ultra-intense laser pulse into a suitable target [101]. The target, a gas jet produced by a gas cell or a supersonic nozzle, forms an “underdense” plasma (i.e., transparent to the laser radiation) upon interaction with the laser pulse, through which the laser propagates. Due to the mass difference between ions and electrons, this force induces a strong, localized, charge separation. When “quasi-resonant” conditions are met, this separation results in a density perturbation that travels in the wake of the laser pulse. The resulting longitudinal EM field is named *Wakefield* and propagates in the plasma at a velocity close to *c*. Due to the longitudinal nature of the electric field (i.e., parallel to the wave propagation direction), electrons can become “trapped" in the accelerating phase of the field and then boosted to relativistic speeds.

Given that the accelerating field in a plasma wave is up to 3–4 times larger than the corresponding field in an RF linac cavity, the most striking feature of laser Wakefield acceleration (LWFA) is its ability to accelerate relativistic energy beams over very short distances. For instance, VHEE beams can be obtained with accelerating structures (i.e., plasmas) on the order of (sub)millimeters. The laser pulse, Wakefield, and accelerated electron bunch are depicted in Figure 7.

In order to be effectively accelerated by the Wakefield, electrons need sufficient initial energy. LWFA techniques vary in how electrons are injected and trapped within the accelerating region of the traveling electric field. The choice of the injection technique is crucial, as it impacts the beam quality in terms of energy distribution, charge, divergence and, ultimately, emittance. While a comprehensive description of all injection techniques is beyond the scope of this review, readers are directed to [103] for further details.

In the context of RT applications, the main features of LWFA-driven VHEE accelerators have been discussed in detail [14,75] and are summarized below:*Beam pulse duration*: LWFA electron bunches inherently feature durations ranging from a few fs up to ps; these are several orders of magnitude shorter than those of medical RF linacs (μs). Similar pulse durations can be obtained with advanced research-type RF linacs. Since the charge per pulse is comparable, this results in a much higher peak dose–rate in the LWFA compared to medical RF linac bunches, enabling UHDR investigations [104].*System compactness*: Unlike RF cavities, plasma can sustain extremely high electron fields, up to 100 GV/m or more for plasma electron densities of $n_{e}$ = $10^{19}$
${\mathrm{cm}}^{-3}$ [105]. In comparison, the electric field in RF cavities is typically around 10–100 MV/m. Therefore, LWFA can boost electron energy over much smaller distances, making compact clinical installations possible [104].*Flexible beam parameters*: The properties of LWFA-driven VHEE beams, such as energy and charge per bunch, can be adjusted by modifying the laser parameters and/or the target gas geometry, composition, and density [106,107].*Beam steering*: The VHEE beam direction follows mostly the laser propagation axis, which allows for active scanning techniques to be implemented by moving the final focusing optics and the gas target [13].*System safety*: No overall shielding of the accelerator system is needed, as the laser propagates from the laser facility up to the gas-jet target. Shielding is required only for the small section housing the accelerating structure [13].

Conversely, several issues with LWFA-driven VHEE accelerators still need to be addressed.
*Repetition rate*: The current repetition rate of LWFA-driven VHEE accelerators is limited by the low repetition rate of high-power lasers (typically 10 Hz), which restricts the mean dose–rate and/or the beam size. Indeed, mean dose–rates of Gy/min have been reported in recent literature [76]. Laser systems with repetition rates of 100 Hz and above are highly desirable, as they are expected to increase the mean dose–rate by 1–2 orders of magnitude [108]. Driven by research infrastructure developments [109,110], such systems are emerging at an industrial level and will likely become operational in a few years.*Stability*: A critical feature hindering the forthcoming clinical use of LWFA VHEE beams is their poor stability, characterized by shot-to-shot fluctuations of bunch parameters, mainly energy, charge, and beam-pointing [111]. Recently, major progress has been made in this field introducing correlation studies of parameters influencing the electron beam stability [112,113]. Similar to the mean dose–rate, beam stability could be enhanced by high repetition rate lasers [114] introducing active feedback control loops. Furthermore, positive results in reducing beam-pointing fluctuations have been obtained using a magnetic beam control system [15,77].*Beam energy*: Unlike RF linac-based VHEE beams, LWFA beams produced using injection mechanisms, are characterized by a broad to moderate electron energy spectrum, with the low-energy end (below 40 MeV) typically eliminated by the beam transport system [15,77]. This issue can affect beam focusing and must be accounted for in dosimetric considerations.

#### VHEE Laser-Driven Accelerators: Overview of Studies

The use of laser–plasma accelerators for RT applications was first envisaged by Glinec et al. [13] and later theorized by Malka et al. [104]. In their study, Glinec et al. investigated the dosimetric aspects of a VHEE laser-accelerated electron beam. The beam was generated at the Laboratoire d’Optique Appliquée (LOA), Salle Jaune facility [115] (CNRS/École Polytechnique/l’ENSTA-Paris). The beam specifications obtained were then imported into Geant4 MC simulations to model its propagation in a water phantom, assuming a 10 mrad FWHM divergence. No air scattering was considered and the beam was propagated in vacuum up to the phantom. The modeled dose distribution in water was considered relevant for RT applications. Due to the quasi-monoenergetic nature of the beam, with a 170 MeV peak and 40 MeV FWHM energy spread, the shape of the dose distribution in the first 10 cm of the water phantom was found to be strongly dependent on the initial electron distribution. This effect, however, diminishes at greater depths where scattering dominates. Fuchs et al. [21] and Lundh et al. [105] continued the work of Glinec et al. at LOA. In the experiment conducted by Semushin and Malka [116], which served as the basis for the study by Fuchs et al., laser-driven beams were generated with peak energies from 60 to 250 MeV. In the study by Fuchs et al., beams with peak energies of 150, 185, and 250 MeV, and corresponding δE/E energy spreads of 11.5, 8, and 6.5%, respectively, were imported into Geant4 to investigate the influence of laser-driven characteristics, primarily beam energy, energy spread, and angular spread, on dose distribution. Furthermore, using the same MC toolkit, a prostate IM-VHEET treatment was simulated, considering the beams with energies of 150 and 250 MeV. In the work by Lundh et al., a quasi-monoenergetic laser pulse was generated at LOA using externally injected LWFA: a 120 MeV peak with a 20 MeV energy spread. Simulated and measured dose depth distributions, the latter obtained using the Geant4 MC toolkit, were compared within a water phantom. VHEE beams longitudinal profiles were found to be suitable for efficiently covering deep-seated targets.

Subiel et al. [31] assessed the response of radiochromic film, a common secondary dosimeter, to laser-driven VHEE beams. Film irradiation was performed at the advanced laser–plasma high-energy accelerators toward X-rays (ALPHA-X) facility [117] within the Terahertz to Optical Pulse Source (TOPS) at the University of Strathclyde, UK [118], using a VHEE beam spectrum of 135 ± 44 MeV (rms).

More than a decade later, Kim et al. [78], at the Korea Electrotechnology Research Institute (KERI) [119], evaluated the dosimetric and physical properties of VHEE beams. The beam spectrum peaked at 94 MeV with an 80 MeV σ. Labate et al. [16] demonstrated that complex anatomical scenarios could be successfully irradiated using LWFA VHEE beams, utilizing stereotactic treatments and intensity modulation techniques similar to those characterizing photon IMRT. This study was carried out using the 220 TW laser beamline at the Intense Laser Irradiation Laboratory (ILIL) [120] at CNR, INO (Pisa, Italy). The beam spectrum ranged approximately from 50 to 250 MeV, with a relatively flat profile above 100 MeV.

Svendsen et al. [15], the Lund High-Power Laser Facility [121] (Lund, Sweden), and Guo et al. [77], Tsinghua University (Beijing, China), demonstrated that magnetic beam control leads to a reduction in beam dimensions and beam-pointing instability. For example, according to Svendsen et al., when quadrupoles are in place, the beam-pointing instability is reduced by an order of magnitude, while the beam spatial profile decreases threefold along the horizontal axis and twofold along the vertical axis. In these studies, the very broad beam spectrum spanned from 0 to 150 [15]/175 MeV [77], with a peak at 95 MeV in [15], and a flat distribution in [77]. Additionally, Glinec et al., Svendsen et al., and Guo et al. demonstrated that by combining LWFA with a magnetic control system, a deeply penetrating VHEE beam can be generated, yielding excellent 3D dose distributions. As will be described in Section 6.2, all these authors consider a magnetic system, comprising three magnetic quadrupoles, for VHEE beam focusing. On the other hand, Zhou et al. [23] (the same group as Guo et al.) presented a new focusing system, based on two magnetic dipoles, specifically designed for LWFA beams.

Significant efforts are needed to transition LWFA structures from research platforms to treatment machines. Guo et al. [77] presented the first stable and compact LWFA treatment set-up, allowing the delivery of homogeneous dose distributions. The designed machine, about 2.8 × 1.4 $m^{2}$ in size, could, potentially, be easily installed within a hospital-based therapy room. Stable operations of a prototype were assessed across 22 weekdays, with 2000–3000 shots per day, equating to tens of hours of continuous running per day over the course of a month. The prototype was used for radiobiological assessments, including mice irradiation, with the aim of comparing VHEET to photon RT.

LWFA VHEE systems, in conjunction with a robotic arm, represent another safe, compact, and efficient concept worth developing. In this regard, Nakajima et al. [79] presented the conceptual design of an LWFA-driven VHEET robotic system. Energy-amplified, intense laser pulses are generated by a drive laser system and guided through a vacuum structure within the main robot body to an accelerator chamber attached to the robot’s head. Within the accelerator chamber, laser pulses focus on the gas target, and the accelerated VHEE beam is then transported and delivered to the patient The six degrees of freedom of the robotic arm allow the target to be irradiated from multiple directions.

To develop robust laser-driven VHEE sources, it is noteworthy that the main laser parameters in the cited experiments are typical of commercial systems. An overview of beam parameters, where available, is provided in Table 3.

## 6. VHEE Beam Focusing

Beam focusing enables the concentration of dose deposition at deep locations, offering (1) a lower entrance dose, (2) reduced lateral scattering in depth, and (3) precise targeting of small 3D volumes while sparing organs at risk (OARs) and healthy tissue.Strong VHEE beam focusing is achieved using magnetic quadrupoles, with each one focusing the beam in a transverse plane and defocusing it in the other. The quadrupole’s magnetic field lines are shown in Figure 8, while Figure 9a–c depict graphic renderings from the thin lens approximation and matrix formalism for focusing and defocusing quadrupoles and drifts, respectively. The final focus can be symmetrical (Figure 9d) or asymmetrical (Figure 9e), depending on whether the beam converges in one or both planes. A collimated beam, with only air scattering divergence accounted for, is considered non-focused. The definitions of *asymmetrically* focused/*line* and *symmetrically* focused terms are presented in Table 1. Monochromatic VHEE beams from RF linacs allow for uniform focusing using quadrupoles. In contrast, LWFA-produced VHEE beams have a broad energy spectrum, meaning that only electrons with matching energies will be focused together.

### 6.1. Focusing Monochromatic VHEE Beams from RF Linacs

Using FLUKA MC simulations, Kokurewicz et al. [32] first evaluated the effects of an ideal focusing system for VHEE beams. A 200 MeV VHEE beam with a diameter (D) of 20 cm was modeled, focused at a depth of 15 cm into a water phantom. The source-to-phantom distance, set at 9.1, 12.5, 16.7, 22.3, 30.1, 41.7, 61, and 99.3 cm, plus the focusing depth, yielded the focal (F) length, i.e., the source to focusing depth distance. As defined in the optics, the *f-number = F/D* represents the focusing geometry; strongly focused beams were observed with an *f-number* ≤ 3.8 (*F* = 61 cm), whereas non-focused, collimated beams corresponded to *f-number* = *∞*. The shape of the volumetric element was found to be symmetrical with respect to the propagation axis for all *f-numbers*. At small *f-numbers*, the longitudinal and transverse dimensions of the volumetric element were comparable, with the longitudinal dimension being slightly larger, resulting in a nearly spherical volume shape. By increasing the *f-numbers*, the spot size decreased and the longitudinal dimension of the volumetric element increased, yielding a more elliptical shape. For an *f-number* of 1.2 or 2.8, with respect to the *f-number* = *∞*, the max surface dose was reduced 40 and 211 times, respectively, and the total integral dose was decreased by 77.8 and 84.9%, respectively. For an *f-number* of 11.5, the peak dose at the surface was almost as high as the peak dose, whereas for an *f-number* of 1.2, it was more than 10 times lower. By shifting the source-to-phantom distance, the investigation was repeated to focus depths of 5 and 10 cm into the phantom; it was found that the peak dose decreased significantly with depth.

The first experimental analysis of focused VHEE beams was performed by Kokurewicz et al. [69]. Initially, the VHEE beam was expanded by using three magnetic quadrupoles, followed by focusing on another set of three quadrupoles. Due to technical constraints, only a line focus with a short focal length was achieved on the horizontal axis. For comparison, a symmetrical focus was examined using FLUKA MC simulations. Within a water phantom, the line focus was achieved at a depth of 14 cm for a 158 MeV VHEE beam (*f*/11.2), and at a depth of 10 cm for a 158 MeV VHEE beam (*f*/12.3) and a 210 MeV VHEE beam (*f*/18.2). Compared to a collimated VHEE beam, the line focus reduced the entrance dose. The symmetrical focus, on the other hand, shifted the $d_{max}$ at deeper ranges. This resulted in a more pronounced dose peak, with a reduction of the proximal and distal doses, providing a more favorable dose distribution. However, due to scattering, the peak dose was lower when symmetrical focusing was applied.

Whitmore et al. [122], conducted an MC TOPAS simulation study to compare the dose distributions of symmetrically focused, asymmetrically focused, and collimated 250 MeV VHEE beams. Focusing was achieved using quadrupole magnets, with four magnets employed for line focusing and six for symmetric focusing. The focal point depths for the collimated, asymmetrically focused, and symmetrically focused beams were 4, 12.6, and 17.4 cm, respectively. At the focal point, the Gaussian beams $\sigma_{x}$ and $\sigma_{y}$ were comparable. The symmetrically focused beam did not exhibit sharp focusing at $d_{max}$, whereas the asymmetrically focused beam showed a sharper focus along the horizontal axis. Symmetric focusing had minimal impact on the dose distribution shape, while asymmetric focusing resulted in noticeable differences in beam shape along the two axes. Entrance doses for the collimated, asymmetrically focused, and symmetrically focused beams were 91, 41, and 25% of the peak dose, respectively. At the phantom entrance, the symmetrically focused beam was sharply focused, while the asymmetrically focused beam had a broader spread, leading to a lower dose at any point of the phantom surface. Exit doses were similar across all beam types, with the asymmetrically focused beam showing slightly more spread. Additionally, asymmetrically focused VHEE beams of varying energies (100, 150, 200, and 250 MeV) were studied. Higher beam energies resulted in more sharply focused on-axis dose distributions due to reduced penumbra and increased focusing strength. The focal spot σ for the 100 MeV beam was 55% larger than for the 250 MeV beam. Entrance doses for the 100, 150, 200, and 250 MeV beams were 45, 36, 31, and 25% of the peak dose, respectively, with focal point depths of 14.4, 15.9, 17.1, and 17.4 cm. The σs values for the 200 and 250 MeV beams were less than or equal to 1 cm in both the *x* and *y* directions. Whitmore et al. were pioneers in demonstrating that focused VHEE beams could be combined to create a spread-out electron peak (SOEP), analogous to the spread-out Bragg peak (SOBP) used in proton therapy. By adjusting the final quadrupole strength, the focus of four beams was set at different depths, and these beams were then weighted and combined to produce a flat dose region spanning 5 cm (from 13 to 17 cm).

Krim et al. [123] developed the virtual magnetic approach (VMA) to reduce the lengthy computational times of MC simulations. The VMA, developed with the GATE MC toolkit, is an in silico method that enables fast and efficient VHEE beam focusing by creating a virtual particle source. Each particle’s momentum is adjusted to ensure it is focused, symmetrically or asymmetrically, at a selected focal depth *f*. The depth–dose curves obtained using VMA modulation were compared with those from Whitmore et al. [122], showing a ±2% difference. The spot size of the focused beam was also estimated for different VHEE beam energies, revealing a fivefold decrease in spot size as the beam energy increased from 100 to 200 MeV. Krim et al. highlighted the advantages of VHEE beam focusing for anatomical variations. They simulated a prostate RT plan with focused VHEE beams, demonstrating the technique’s practical benefits in treatment planning.

In their 2024 study, Whitmore et al. [12] experimentally tested a focusing system consisting of six quadrupole magnets using a 201 MeV VHEE beam with a 2 MeV energy spread. The beam was asymmetrically focused, resulting in a wide, non-diverging beam, along the vertical axis and a strongly focused beam along the horizontal axis. The depth–dose curves were reconstructed from measurements taken within a water phantom and were compared with results from MC TOPAS simulations, in both water and air. The in silico evaluation indicated that focusing in water was less effective and occurred at shallower depths compared to air, mainly due to electron scattering. Experimentally, at the focal point in water, the beam’s horizontal $\sigma_{x}$ was reduced to 50% of its initial value. Additionally, the study demonstrated that adjusting the magnetic strength of the final quadrupole allowed the focal point to shift almost linearly, facilitating the first experimental production of SOEP. Due to technical limitations, this evaluation was restricted to focusing depths less than 6 cm, with longer depths being modeled.

### 6.2. Focusing Polychromatic LWFA-Produced VHEE Beams

Inspired by the previous work of Fouktal et al. [124] on proton therapy, Glinec et al. [13] first conceptualized the use of focusing systems in the context of VHEET. By means of MC Geant4 simulations, focusing was achieved via three quadrupole magnets. Before focusing, the LWFA-derived VHEE beam, retrieved from experimental tests, was filtered by a monochromator to reduce its energy width. Thanks to beam focusing, the authors observed several dosimetric improvements. First, $D_{max}$ was higher, with $d_{max}$ shifting deeper within the phantom. Second, the electron fluence was more concentrated around the central axis; thus, compared to collimated beams, the transverse and lateral dose distributions decreased more slowly. Third, within the first 6 cm, focusing compensated for beam broadening due to electron scattering. Consequently, isodose curves remain parallel to the longitudinal axis, allowing for better sparing of healthy tissues near the entrance channel. The focusing system developed by Glinec et al. was also used by Fuchs et al. [21], who found that VHEE beam focusing improves lateral penumbra. This technique was also applied in a prostate RT plan.

The first experimental assessment of LWFA-based focused VHEE beams was performed by Svendsen et al. [15]. The beam had a maximum energy of 160 MeV and a 90 MeV peak. To focus the beam, three quadrupoles were considered. Given the beam spectrum, their focal points were determined at an energy of 90 MeV. By varying the focal position within the phantom, the depth-dose profile was tuned. The penumbra was smaller when the VHEE beam was focused at the entrance and larger when focused at the exit. The maximum dose was always located at the entrance except when the beam was focused 3 cm behind, in this case, $d_{max}$ was 7.2 mm. In general, being the VHEE beam polychromatic and the focal point set for 90 MeV, if the focus was at the exit, all electrons entered the phantom converging. Beam electrons with energies lower than 90 MeV focus at shallow depths, while those with energies higher than 90 MeV focus at greater depths Typically, low-energy electrons are more scattered; however, in this specific situation, low-energy electrons are also more strongly focused. Thus, by focusing the beam at the phantom exit, the scatter is reduced. For this reason, when the beam is focused at the exit, the central-to-final end of the phantom receives the highest dose. Conversely, by focusing the beam at the phantom entrance, all electrons whose energies are lower than 90 MeV will diverge, resulting in a shallower depth dose curve.

Zhou et al. [23] pointed out that quadrupole magnets are unsuitable for beams with large energy dispersion and proposed a two-dipole focusing system. The focusing distance *f*, calculated from the second dipole’s front face, depends solely on the magnets’ strengths and lengths. Using TOPAS MC simulations, several VHEE beams were modeled to reflect those from standard LWFA beamlines. In different runs, the beams had an energy of 200 MeV, with rms values of 0.1, 5, and 20%, and a uniform energy distribution between 100 and 300 MeV. The dipole parameters were set for a beam focusing at *f* = 15 cm. The beam was asymmetrical, with electrons at the phantom entrance converging along the horizontal axis and diverging along the vertical axis. Despite significant disparities in energy spread, the system could focus all VHEE beams, with comparable dose distributions and dose peaks concentrated at similar depths. The VHEE beams with a 200 MeV peak energy exhibited negligible differences. The beam with a 100–200 MeV uniform energy distribution showed minor variations of less than 5%. By modifying the magnetic strengths, the focal depth was set to increase from 7 to 20 cm. The surface dose, estimated for a 300 MeV beam with 5% rms, slightly increased with the focal depth, rising from approximately 26% for *f* = 7 cm, and to 56%, for *f* = 20 cm. Lastly, similar to the work by Whitmore et al. [122], five- and seven-weighted VHEE beams, each associated with different dipole magnetic fields, were summed together to provide a flat high-dose plateau of 5 cm (from 8 to 13 cm) and 7 cm (from 8 to 17 cm) with σ values of 0.45 and 0.34%, respectively.

## 7. Secondary Dosimetry for VHEE Beams

Establishing accurate dosimetry protocols with standard secondary dosimeters is crucial for the clinical translation of VHEET in hospitals. Both radiochromic film and ionization chambers, well-established for low-energy electrons, have been considered for VHEET. While VHEE beams at CONV dose rates pose dosimetric challenges, the FLASH effect further complicates finding the optimal instrument.

Radiochromic film is commonly employed as a self-developing dosimeter, with well-documented properties for photons, protons, and low-energy electrons [125,126]. Films are tissue-equivalent with high spatial resolution; they exhibit energy independence in the 6–20 MeV range [125], and dose–rate independence of up to 1.5·$10^{10}$ Gy/s [127]. However, films require a ∼24–48 h post-irradiation process, making them less suitable for clinical tests where a beam monitoring system with real-time readout would be preferred.

Ionization chambers, the RT gold standard, are the most practical online clinical dosimeters. However, they suffer from high recombination loss, and initial calibration against a primary standard is not available for VHEE beams. VHEE beams come in ultra-short fs/ps pulses, $10^{6}$–$10^{8}$ times shorter than those of clinical linacs. To this extent, Subiel et al. [128] evaluated the energy downshift of a 150 MeV VHEE beam as it penetrates a water phantom using Geant4 and FLUKA MC simulations. At a 17.5 cm depth, the peak energy was reduced to 112 MeV. As the bunch length increases with beam broadening, it was raised from ∼1.1 fs, at the phantom entrance, to 1 ps, at a 30 cm depth (several orders of magnitude shorter than clinical linacs).

The search for the most suitable VHEET dosimeter began with systems already in clinical use. However, new VHEET-tailored prototypes show promising potential for both CONV and FLASH regiments.

### 7.1. Radiochromic Film

Subiel et al. [31] performed dosimetric measurements with VHEE beams on a water phantom with ten films. Beams were generated at the ALPHA-X laser–plasma accelerator, 135 MeV, and at the SPARC RF linac, 165 MeV. Bazalova-Carter et al. [71] considered nine films within a polystyrene phantom; the set-up was impinged with 50 and 70 MeV VHEE beams at NLCTA. In both studies, results were benchmarked with FLUKA [31] and EGSnrc [71] MC simulations. The film calibration curves were based on the measured responses at 20 MeV (Clinac iX, Varian) [31] and 12 MeV (Trilogy, Varian) [71]. For both the laser and RF linac-generated beams, Subiel et al. found the experimental and in silico measured dose profiles along the central beam axis to be in excellent agreement. Similarly, Bazalova-Carter et al. reported good agreement between experimental and simulated data; the FWHM and dose discrepancy values were found to be, at most, 4 and 5%, respectively. Both Subiel et al. and Bazalova-Carter et al. investigated film response in the VHEE energy range. Previous proton therapy studies suggested a potential film response dependence on LET [129]. For this reason, Subiel et al. simulated the LET spectrum for the 20 MeV electron beam used in calibration as well as for the ALPHA-X beam (135 MeV peak) at a few depths within the phantom. At all energy values, a very similar LET spectrum was observed, leading to a negligible film response dependence on LET and, in turn, on beam energy. Subsequently, based on the model from Sutherland and Rogers [130], Bazalova-Carter et al. modeled the film response to electron beams, 1 to 100 MeV in energy, and reported a flat ≤2.5% energy response. Bazalova-Carter et al. further experimentally studied the film response of a 60 MeV VHEE beam at a dose–rate of ∼$10^{12}$ Gy/s. For this assessment, the polystyrene phantom was irradiated with 3, 2, and 1 pulses at 3 × $10^{12}$, 4.5 × $10^{12}$, and 9 × $10^{12}$ Gy/s, respectively. Results were dose–rate independent within 3.7 ± 3.5%.

Lagzda et al. [24] and Clements et al. [70] used films at CLEAR in the framework of their VHEET investigations. In both studies, experimental results were compared with the outcomes of TOPAS MC simulations. Film calibration was performed with 15 MeV (commercial Elekta linac) [24] and 5.5 MeV electrons (eRT6 Oriatron) [70]. Lagzda et al. evaluated VHEE beam dose penetration within heterogeneous media. Experimental tests were performed with a 156 MeV VHEE beam on a water phantom with ten films within. Results were consistent with the simulation within a 5% uncertainty. Clements et al. considered VHEE GRID therapy and performed experimental tests with and without a mini-GRID collimator. In all cases, a VHEE beam with energies of 140, 175, and 200 MeV was directed onto a water phantom with films inserted. In silico modeling was performed to reproduce the 200 MeV VHEE beam run. Considering the open beam set-up, the $\sigma_{x}$ and $\sigma_{y}$ differences between MC and film doses were below 0.5% on average. The central axis dose was 3% on average, and worse at a depth of up to 30 cm. In the mini-GRID set-up, the peak-to-valley-dose ratio was lower by 14%, while the central axis dose and the valley dose were 14 and 27% higher on films. The high central axis and valley doses could be caused by the Bremm. contamination at the collimator, as captured by the films.

### 7.2. Ionization Chambers

Low-energy electron dosimetry at UHDR via ionization chambers was studied by Petersson et al. [131], Jaccard et al. [48], and Jorge et al. [132], using an Oriatron eRT6 linac. Petersson et al. showed recombination losses of up to 70%.

Typically, with low-energy electrons and CONV dose–rates, ion recombination in the ionization chamber is addressed with the standard two-voltage analysis (TVA) [133,134,135]. Several theoretical models of ion collection efficiency in ionization chambers have been presented by Boag et al. [136,137] and Burns and McEwen [138]. Di Martino et al. [139] developed a recombination model specifically designed for IORT, while Petersson et al. [131] proposed an empirical model by fitting a logistic function to the data. Finally, Di Martino et al. [140] showed a new calculation method for the UHDR regimen.

Using the TVA, Subiel et al. [128] estimated the IBA CCO4 chamber recombination factor $f_{ion}$ for a Varian iX linac 20 MeV beam and a 165 MeV beam from SPARC. The $V_{1}$/$V_{2}$ ratio was 2 or 3, with $V_{1}$ equal to 300 V and $V_{2}$ at either 150 or 100 V. At $V_{2}$ values of 150 and 100 V, respectively, the $f_{ion}$ values were 1.0100 and 1.0094 for the 20 MeV electron beam, and 1.5953 and 1.5968 for the VHEE beam, indicating a general 60% recombination.

McManus et al. [67] and Poppinga et al. [66] performed charge measurements with ionization chambers and compared them to the dose readings of different dosimeters. Both studies were performed at CLEAR with a 200 MeV VHEE beam. McManus et al. used a PTW Roos Type-34001 ionization chamber, with voltages of 75, 200, 350, or 600 V, at a dose–rate per pulse ranging from 0.03 and 5.04 Gy/pulse. In parallel, graphite calorimetry from the National Physical Laboratory, the UK’s primary standard, was used. The $f_{ion}$ value was evaluated by comparing the absolute dose readings from the chamber and the calorimeter. The previous findings of Subiel et al. were confirmed: collection efficiency increased with voltage and markedly decreased as the pulse dose–rate increased. The chamber collection efficiency, similar to those from clinical linacs, ranged from up to 97% (0.03 Gy/pulse) to 4% (5.04 Gy/pulse). Furthermore, the Boag [137], Di Martino [139], and Petersson [131] recombination models, as well as the TVA method, were applied to analytically esteem the $f_{ion}$ value, using the same voltages and pulse dose–rates as the experimental results. The Boag and Di Martino models, including the free-electron fraction (i.e., the fraction of charge, which originates from the collection of free-electrons instead of negative ions), were found to fit the experimental data reasonably well, although the $f_{ion}$ value was not accurately predicted over the whole pulse dose–rate range. The TVA method significantly underestimated $f_{ion}$ for pulse dose–rates higher than 0.5 Gy/pulse. Finally, the logistic model from Petersson yielded the most accurate $f_{ion}$ esteem, with accurate predictions over the full pulse dose–rate range. Poppinga et al. considered a PTW 34045–Advanced Markus ionization chamber, with a voltage of 400 V, at a dose–rate per pulse ranging from 0.2 to 12 Gy/pulse. In parallel, 15 films, calibrated with a 15 MeV electron beam (Siemens Primus), were employed. Both the chamber and films were inserted in a water phantom. The $f_{ion}$ from the chamber/film dose readings comparison was retrieved at different depths. At the highest pulse dose–rate, a 30% collection efficiency was observed. No significant change in ion collection efficiency was reported by doubling the beam size from 3.5 to 7 cm.

### 7.3. Novel Concepts

Novel dosimetry devices have recently been proposed for low-energy electron beams at UHDR. In this framework, silicon [141,142], diamonds [143] and multi-layered nanoporous aerogel high-energy-current [144] state solid detectors have been introduced. Regarding VHEET, new beam monitoring concepts were investigated by Bateman et al. [64], Hart et al. [18], and Rieker et al. [97], with potential use specifically suited for FLASH-RT. These new technologies provide online monitoring and their responses do not saturate at ultra-short beam pulses, even at UHDR. All studies took place at CLEAR, with beam energies of 160 [64] and 200 MeV [18,64,97]; results were benchmarked against film dosimetry. Bateman et al. presented the fiber optic flash monitor (FOFM), which comprises an array of silica optical fiber-based sensors. In this system, the Cherenkov signal is produced upon radiation and coupled with a photodetector (CMOS camera) for signal readout. Initial results showed that the FOFM beam monitor exhibited a linear response from 0.9 to 57.4 Gy/pulse. No beam energy or instantaneous dose–rate dependence was observed. The authors emphasized the possibility of pulse-by-pulse beam size measurements, a potentially crucial feature in FLASH-RT, currently non-available with ionization chambers. The use of plastic scintillator detectors (PSDs) as dosimeters for VHEET at UHDR was investigated by Hart et al. PSDs present many advantages: online dose readings, dose–rate independent responses, sub-millimetric spatial resolutions, and radiographic properties similar to those of human tissue. Two PSDs were considered: a polystyrene-based (BCF-12) and a proprietary polyvinyltoluene-based material (PTV). Irradiation demonstrated linearity for dose–rates up to 1.16 × $10^{9}$ and 9.92 × $10^{8}$ Gy/s within a single pulse, for BCF-12 and PTV, respectively. Nevertheless, accumulated doses on the order of kGy may diminish the light output, necessitating routine recalibration. Lastly, the authors highlighted the excellent radiation hardness of the detectors, with output decreasing by less than 1.5%/kGy at 200 MeV. They noted that damages of up to 18 kGy were temporarily recoverable, but permanent damage occurred at higher doses. Reiker et al. presented a dosimetry method using cerium-activated yttrium aluminum garnet (YAG:Ce) scintillating crystals positioned perpendicular to the beam. The scintillation light was reflected by a mirror located downstream of the scintillator and directed toward a vertically displaced digital camera. The measured beam sizes and their longitudinal evolution in air showed good agreement between YAG and radiochromic film readings, aside from a systematic YAG underestimation of 10% and an overestimation of 5% in CONV and FLASH irradiation, respectively. While the cause of these offsets is not yet fully understood, they may be associated with the YAG signal-to-noise ratio, which increases in FLASH mode due to the higher dose per pulse.

## 8. Radiobiology of VHEE Beams

### 8.1. The Estimation of RBE for VHEE Beams

Relative biological effectiveness (RBE) is defined as the ratio of the dose of one type of radiation relative to the dose of a reference radiation, typically Co60 X-rays, yielding the same effect. Many studies have determined the RBE of low-energy electrons (≤50 MeV). Specifically, electron RBE was quantified for beam energy values of 6 [68], 10 [68,145], 11 [146], 15 [68], and 50 MeV [147]. All RBE values were ∼1.

Considering 100, 150, and 200 MeV VHEE beams, Small et al. [68] used the CLEAR linear accelerator at CERN to perform the first irradiation of pBR322 plasmid (i.e., DNA ring-like structures found in bacteria [148]), carried out in both aqueous and dry environments. Very high yields of single- and double-strand breaks (SSBs and DSBs) were observed post-VHEE beam delivery. Thus, with damages caused in more than 99% of cases, it was inferred that this type of radiation primarily induces DNA damage through indirect effects. Specifically, after the radiation-induced dissociation of water molecules (radiolysis of water), free radicals (mainly ${\mathrm{OH}}^{-}$ and $H^{+}$) rupture the DNA helix [149]. Furthermore, due to the small variation in linear energy transfer (LET) (0.220–0.226 keV/μm), only a slight difference in DSB yield was observed over the 100–200 MeV energy range. Lastly, DSB yields were used as biological endpoints for RBE calculation [150]. The RBE of VHEE was estimated to be 1.1–1.2 for aqueous plasmids and almost 1 for dry plasmids. A potential RBE increase with electron energy was hypothesized.

Delorme et al. [151] compared the macro- and micro-dosimetric properties of Co60 X-rays, 20 MeV electrons, 100–300 MeV VHEEs, 154 MeV/n carbon ions, and 262 MeV/n neon ions. The evaluation was performed using the MC GATE toolkit with a numerical approach. The macroscopic metric was the dose-averaged LET ($\bar{L_{d}}$), while the microscopic metrics were the dose mean linear energy ($\bar{y_{d}}$) and the dose-weighted linear energy distribution (yd(y)). From a macro-dosimetric point of view, the $\bar{L_{d}}$ ratio values of 300 MeV VHEEs compared to protons, 100 MeV VHEEs, and 20 MeV electrons, were 0.2, 1.9, and 3.3, respectively, positioning VHEEs somewhere between low-energy electrons and protons in terms of biological efficiency. Conversely, the micro-dosimetric data revealed no substantial difference between VHEEs and low-energy electrons with indistinguishable survival curves. VHEE RBE was estimated to be ∼1, using survival curves as the biological endpoint.

Wanstall et al. [19] experimentally (ARES, DASY) delivered a 300 kVp X-ray beam and a 154 MeV VHEE beam to prostate (PC3) and lung (A549) cells in suspension. A clonogenic assay was performed to determine the VHEE beam RBE values in cancer cells. The evidence suggested that VHEEs are as damaging as photons. The RBE values were quantified at 50 % ($D_{0.5}$) and 10% ($D_{0.1}$) cell survival: for PC3 cells, the results were 0.74 ($D_{0.5}$) and 0.93 ($D_{0.1}$), while for A549 cells, the results were 0.93 ($D_{0.5}$) and 0.95 ($D_{0.1}$).

Based at INFLPR (Magurele, Romania), Orobeti et al. [63] carried out experimental tests using high-intensity laser–plasma VHEE beams performed on A375, radiation-resistant human metastatic melanoma cells, and NHEM (normal human epidermal melanocyte) co-cultures, both grown on chamber slides. The used VHEE beam was polyenergetic, with a ∼190 ± 40 MeV peak energy. Serving as positive controls for radiation-induced DSBs, in parallel tests, cell co-cultures of the same type were irradiated with pulsed X-rays. By means of a microscopic p-γ-H2AX foci count, the DNA damage foci in response to radiation stress were quantified. The occurrence of DNA damage foci is the first consequence of radiation damage inflicted on the targeted cells [152], and the p-γ-H2AX count represents a marker for DSB formation. Interestingly, the number of induced foci, after 30 min of incubation at 37 °C following beam delivery, was higher with VHEEs than with photons, at a cumulative VHEE beam dose one order of magnitude lower than that of photons. This significant finding warrants further investigation. To date, the study by Orobeti et al. is the first radiobiological study performed on laser-driven VHEE beams. While a few studies have focused on the genetic effects of laser-driven electron beams ([59,153,154,155,156]), all of these studies have used low-energy electron beams and various biological endpoints.

### 8.2. VHEE Radiobiology and FLASH

Small et al. [68] and Wanstall et al. [62] experimentally investigated the dose–rate impact on the irradiation of pBR322 aqueous plasmids using VHEE beams with energies of 100 [68], 150, and 200 MeV [62,68] using CLEAR. Small et al. considered a dose–rate of ∼0.5 Gy/s for CONV-RT and >$10^{8}$ Gy/s for FLASH-RT, while Wanstall et al. examined plasmid DNA damage at dose–rates of 0.08 Gy/s for CONV-RT and 96 and 2 × $10^{9}$ Gy/s for FLASH-RT. Small et al. used the DSB yield as the biological endpoint. At all beam energies, Small et al. did not observe any statistically significant variations in DSB yield with the dose–rate. This result was attributed to the lack of key experimental conditions essential to the FLASH effect, namely the use of oxygenated water and room temperature ( 25 °C). Conversely, Wanstall et al. chose SSBs, whose frequencies were significantly higher than DSBs, as biological endpoints. Additionally, Wanstall et al. increased the hydroxyl radical scavenger Tris concentration to 10 or 100 mM, (whereas Small et al. used a 1 mM concentration). By doing so, the plasmids’ scavenger capacity was markedly increased, reaching the levels typically characterizing cells. Samples were irradiated with the plasmid solution at room temperature/air conditions. With a notable reduction in SSBs from CONV-to-UHDR, the FLASH effect was indeed observed in a cell-like environment after VHEE beam irradiation. For instance, at 200 MeV, the magnitude values of the FLASH sparing quantified in the SSB reduction were 27 and 16% in the 10 and 100 mM Tris environments, respectively.

Orobeti et al. [63] (INFLPR) exposed A375 melanoma tumor cells and NHEM normal melanocyte cells to VHEE UHDR irradiation with beam energy peaking at 190 MeV. The radiation impact was lower on the non-cancerous NHEM cells.

These studies show, for the first time, preliminary evidence of a differential in vitro response from UHDR to CONV VHEE irradiation.

## 9. VHEET Case Studies

This section begins with an overview of the current status of the development of VHEET treatment planning systems (TPSs), followed by a review of published VHEE clinical studies, where dosimetric calculations were performed using MC toolkits. Investigations primarily focus on IM-VHEET for prostate cancer, although studies also include lung, pediatric brain, non-pediatric intracranial and head, and gastrointestinal tumors. A summary of these studies is provided in Table 4, Table 5 and Table 6. The 3D-CRT (3D conformal radiation therapy) RT modality was also implemented in a few investigations. In general, most studies investigate VHEET-RT treatments by comparing them with photon RT, with fewer comparisons made to proton RT. The quality of VHEET plans is typically assessed by evaluating (1) the integral dose, (2) the target or GTV/PTV (gross/planning target volume) coverage, and (3) the sparing of OARs and external volumes, (i.e., healthy tissues outside the OARs). Furthermore, in a few studies, Table 7, FLASH-RT has been considered.

### 9.1. Treatment Planning Systems for VHEET

A clinical TPS should provide realistic dose distributions within patient anatomies [17]. Currently, with no existing VHEE RT devices, medically certified software is unavailable for VHEET planning [157]. VHEET also lacks a therapeutic protocol to guide clinicians in choosing irradiation geometry, leading most early planning studies to adopt photon plan orientations [157]. To date, VHEET treatment planning studies have exclusively utilized MC simulations [158], with toolkits such as PENELOPE [159], FLUKA [160], EGSnrc/DOSXYZnrc [161], Geant4 [162], TOPAS [163], and/or GATE [164]. These studies have shown that MC codes are accurate for initial VHEET planning [17]. However, within the VHEET energy range, some basic physical quantities are still affected by cross-section uncertainties [17,165]. For instance, the radiative stopping power uncertainty for VHEEs above 50 MeV is approximately 2% [17,165].

Few studies have investigated the new concepts for analytical VHEE TPSs. Ronga et al. [166] and Sitarz et al. [167], from the same group, presented a deterministic algorithm based on the Fermi–Eyges theory of multiple Coulomb scattering [168], while Naceur et al. [158] extended an existing Boltzmann–Fokker–Plank (BFP) chain [169] to VHEE. Algorithms were validated in comparison to the TOPAS and Geant4 MC toolkits, respectively.

### 9.2. Prostate Tumors

#### 9.2.1. Beam Number Selection

Yeboah et al. [4] first reported on the impact of the number of VHEE beams on IM-VHEET prostate plans. IM-VHEET treatment plans were designed using 2 to 25 beams, each with an energy of 200 MeV. Except for the two-beam plan, which used an orthogonal arrangement, the beams were evenly spaced around the phantom. Results indicated that a minimum of nine beams is required for an acceptable plan, with target coverage improving as the number of beams reaches 19–21. Above 21 beams, those on opposite sides of the phantom overlap, resulting in higher doses to sensitive structures. A comparison of the 21-beam plan with the 9-beam plan showed a 25% improvement in the target dose and reductions in rectal and bladder doses of 6.3% and 16.1%, respectively. Garnica-Garza [170] evaluated 3D-CRT VHEET plans with 3, 5, and 9 beams, and beam energies ranging from 75 to 250 MeV. The study found that the GTV dose remained invariant regardless of the number of beams. However, PTV coverage worsened as the number of beams decreased. Acceptable PTV coverage was still achievable with a smaller number of beams if beam energies of at least 200 MeV were used. The non-tumoral integral dose was significantly reduced as the number of beams decreased, particularly at higher beam energies. Regarding the rectum, the average dose decreased with an increase in the number of beams. The maximum rectal dose increased with the number of beams for beam energies below 150 MeV, but for higher energies, the maximum rectal dose reached its minimum value, which was independent of the number of beams. Spek et al. [171] considered IM-VHEET plans with 9, 18, and 36 beams, using beam energies within the 100–400 MeV range. Across all beam energies, the PTV dose remained almost unaffected. However, the dose received by OARs decreased significantly as the number of beams increased. This reduction was more pronounced at beam energies of 100–200 MeV compared to 300–400 MeV.

The aforementioned work by Yeboah et al. was the first to evaluate a form of VHEE arc therapy. They implemented a 75-equispaced beam arrangement. Compared to the 21-beam fixed-angle plan, rotation therapy resulted in a more homogeneous target dose, but with an increased integral dose to the rectum, bladder, and normal tissues.

#### 9.2.2. Beam Energy Selection

The impact of VHEE beam energy on IM-VHEET prostate plans has been the subject of several investigations. Yeboah et al. [4] evaluated a nine-beam prostate plan with beam energies of 50, 100, 150, 200, or 250 MeV. Raising the energy from 50 to 150 MeV led to a 23.3% improvement in target dose homogeneity and reductions in rectal, bladder, and normal tissue integral doses by 24.8, 10.9, and 8.4%, respectively. However, Yeboah et al. noted that further increasing the VHEE energy up to 250 MeV had an insignificant effect on these quantities. This finding disagrees with subsequent studies by Fuchs et al. [21], Garnica-Garza [170], Schuler et al. [73], and Spek et al. [171]. Fuchs et al. found that increasing the energy from 150 to 250 MeV reduces the mean dose to the rectum and bladder by 2.4 Gy. Garnica-Garza evaluated 3D-CRT VHEET plans with beam energies of 75, 100, 150, 200, and 250 MeV. At all VHEE beam energies, the non-tumoral integral dose was lower than its prescribed value and decreased slightly with increasing beam energy. Higher VHEE energies resulted in better sparing for the rectum and bladder, while the femur heads experienced the opposite effect. Both GTV and PTV coverage increased with beam energy, with 150 MeV generally being the minimum VHEE energy for an acceptable plan. Schuler et al. observed an enhanced OAR-sparing improvement by increasing the beam energy from 100 to 200 MeV. Lastly, Spek et al. compared IM-VHEET plans with beam energies of 100, 200, 300, and 400 MeV, finding that target and OAR dose metrics improved with higher energies, with more pronounced differences at lower energies.

**Table 4 cancers-17-00181-t004:** Clinical studies applying VHEET treatments for prostate cancer.

Study	VHEE Beam	VHEE Beam	VHEET	Others RT	FLASH?	Dose
	Energy [MeV]	Number	Technique	Modalities		Engine
Yeboah et al. [4]	(1) 50, 100, 150, 200, 250	(1) 9	IM-VHEET	NO	NO	PENELOPE
	(2) 200	(2) 2, 3, 5, 7, 9, 11,				
		13, 15, 19, 21, 25				
	(3) 50–150–250 ^1^	(3) 9, 75 ^2^				
	(4) 50–100–150–200–250 ^1^	(4) 9				
	(5) 50, 100, 150, 200, 250	(5) 75 ^2^				
Yeboah and Sandison [5]	250	9, 11	IM-VHEET	(a) 15 MV IMRT	NO	PENELOPE
				(b) 200 MeV IMPT		
DesRosiers et al. [172]	200	8	3D-CRT VHEET	16 MV 3D-CRT	NO	PENELOPE
Fuchs et al. [21]	150, 250	7	IM-VHEET	6 MV IMRT	NO	Geant4
Moskvin et al. [173]	200	8	3D-CRT VHEET	16 MV 3D-CRT	NO	PENELOPE
Garnica-Garza [170]	75, 100, 150, 200, 250	3, 5, 9	3D-CRT VHEET	15 MV 3D-CRT	NO	PENELOPE
Bazalova-Carter et al. [174]	100	36	IM-VHEET	15 MV VMAT	YES	EGSnrc + RayStation ^3^
Schuler et al. [73]	100, 200	16	IM-VHEET	(a) 6 MV VMAT	YES	EGSnrc + RayStation ^3^
				(b) IMPT		
Sarti et al. [175]	(a) 70–120–130 ^1^	(a) 7	IM-VHEET	10 MV IMRT	YES	FLUKA
	(b) 70–120–130 ^1^, 100–110–130 ^1^	(b) 5				
Spek et al. [171]	100, 200, 300, 400	9, 18, 36	IM-VHEET	VMAT	NO	FLUKA
Bohlen et al. [25]	50, 100, 150, 200, 250	7	TR-VHEET ^4^	250 MeV TR-PT ^4^	YES	RayStation ^3^

Studies listed in chronological order. ^1^ Energy modulation. ^2^ Simulating arc-therapy. ^3^ Research version. ^4^ Transmission (TR) planning: pencil beam-scanning without intensity modulation.

IM-VHEET plans are typically reported using monoenergetic beams. Yeboah et al. evaluated energy modulation effects in both 9-beam and 75-beam arrangements. For the nine-beam plan, minor improvements in dose distributions were obtained when modulation was performed using no more than three energy bins (50, 150, and 250 MeV). Energy modulation in rotation therapy did not affect target coverage but increased the integral doses to the rectum, bladder, and external tissues by 2.6, 3.9, and 0.9%, respectively.

#### 9.2.3. Comparison to Other Radiotherapy Modalities

##### VHEET vs. Photons

Many studies have compared VHEET with photon RT modalities, including 3D-CRT [3,170,173], IMRT [5,21], and VMAT (volumetric modulated arc therapy) [73,174]. These studies agree that IM-VHEET offers more conformal target dose distribution and comparable or better target coverage than photons.

VHEET provides greater dose-sparing for sensitive OARs, mainly the rectum, bladder, and surrounding external volumes. Moskin et al. [173], DesRosiers et al. [172], and Garnica-Garza [170] compared 3D-CRT plans with VHEE and photon beams. According to Moskin et al., VHEE beams offered more uniform target coverage and a lower dose to normal tissues. DesRosiers et al. found photon 3D-CRT to offer a slightly better rectum sparing, while 3D-CRT VHEET significantly improved bladder sparing. In general, these authors found that VHEE beams moderately outperformed photons in the high-dose range for external volumes. Garnica-Garza found photon dose metrics to be slightly superior for the rectal wall, while the difference between photons and VHEE beams was negligible for the bladder. For the non-tumoral integral volume, Garnica-Garza stressed the superiority of VHEET over photons, with a more targeted dose distribution.

With respect to IMRT, Yeboah and Sandison [5] found that the IM-VHEET mean doses to the rectum and bladder were reduced by up to 10.4 and 10.1%, respectively, while the integral doses to the external volumes were lowered by up to 11.8%. Additionally, they reported a posterior rectal wall dose reduction of up to 19.4% with IM-VHEET, potentially allowing for higher dose escalations with a lower probability of complications. For the rectum and bladder, Fuchs et al. [21] showed that IM-VHEET reduced the minimum dose to 0 and lowered the mean dose by 5 Gy compared to IMRT. Beam focusing led to additional improvements; notably, a reduction of 6.5 Gy in the mean bladder dose was achieved when a 250 MeV beam was focused in a vacuum with a 30 cm focusing distance. Bazalova-Carter et al. [174] found that the IM-VHEET dose values to the rectum and other OARs were 3 and 14–84% lower than those of VMAT, respectively. Conversely, the mean dose to the urethra and penile bulb was 3% higher. For the 100 MeV IM-VHEET plan, Schuler et al. [73] found that the mean doses to the bladder and rectum were 21% and 8% higher, respectively, compared to those of a VMAT plan. However, a 200 MeV IM-VHEET plan demonstrated reductions in the mean doses to the bladder and rectum by 1% and 13%, respectively, compared to VMAT. Similarly, Spek et al. [171] observed that a 100 MeV IM-VHEET plan resulted in higher OAR mean doses than VMAT, with reductions in the OAR mean doses seen only with VHEE beams of at least 300 MeV and more than 18 beams. Regarding the femurs, Fuchs et al. found no dose metric improvements from IM-VHEET plans compared to photons, even with focusing. In their study, the maximum dose to the femurs with IM-VHEET was 7 Gy higher than that with IMRT. Similarly, Spek et al. observed that even in the best IM-VHEET treatment scenario (300 MeV and 18 beams), the femur dose metrics were higher than those of VMAT. According to Garnica-Garza, the percentage volume of the femoral heads receiving a dose above 10% of the prescribed value was significantly lower with VHEET compared to photons. Schuler et al. observed that a 200 MeV IM-VHEET plan reduced the femur dose by 23–27% compared to VMAT. Both Garnica-Garza and Schuler et al. claimed that lower VHEE energy implies reduced OAR sparing for the femoral heads. Schuler et al. reported a femur dose escalation of 13–15% when the VHEE dose increased from 100 to 200 MeV.

##### VHEET vs. PT

Yeboah and Sandison [5] and Schuler et al. [73] both compared IM-VHEET and IMPT (intensity-modulated proton therapy), with IMPT clearly outperforming IM-VHEET. Both studies considered a two-beam IMPT arrangement with beam orientations of $90^{\circ }$ and $270^{\circ }$. For the most complex scenarios, Yeboah and Sandison also considered a four-beam IMPT plan ($0^{\circ }$, $90^{\circ }$, $180^{\circ }$, $270^{\circ }$). IMPT always resulted in better dose homogeneity and a more conformal dose distribution around the target. In addition, IMPT significantly reduced the dose to sensitive structures and external volumes. Compared to IM-VHEET, Yeboah and Sandison found that IMPT delivers a 16.8% lower mean dose to the rectum and bladder and a 23.4% lower integral dose to the external volumes. Schuler et al. reported the mean integral dose to be about 50% lower in IMPT than in IM-VHEET. Both modalities guaranteed efficient sparing of the posterior rectal wall. However, in cases of non-overlapping targets and sensitive structures, IMPT may offer a greater protection of the anterior rectal wall. Based on these results, the authors claim a significant improvement with IMPT compared to IM-VHEET. On the other hand, proton range uncertainties, potentially leading to over-/under-exposure of the distally located OARs, are not considered. More recently, Böhlen et al. [25] compared proton and VHEET plans using the transmission modality (see Section 9.7.3). They utilized a research version of the RayStation TPS for proton pencil beam scanning, which included a newly developed dose engine for VHEET. Seven equispaced co-planar beams were used. Proton plans used an energy value of 250 MeV, while VHEET plans ranged from 50 to 250 MeV in steps of 50 MeV. For deep-seated targets, like the prostate, VHEE beam energies lower than 150 MeV produce large beam penumbra that substantially degrade conformity compared to PT plans. Thus, to achieve similar conformity to PT, VHEET requires an energy value of at least 250 MeV. Although having partially reduced dosimetric conformity, VHEET energies of 150 MeV and above are enough to achieve dose metrics comparable to, although slightly worse than, PT plans.

### 9.3. Lung Tumors

Zhang et al. [176] evaluated an IM-VHEET lung plan with 11 VHEE beams at energies of 100, 120, or 140 MeV, finding that the 140 MeV plan provided the best quality. The first IM-VHEET/VMAT comparative study for lung cancer, presented by Bazalova-Carter et al. [174], found that a 36-beam, 100 MeV IM-VHEET plan may offer superior conformity compared to VMAT. Subsequent studies by the same group, Palma et al. [72] and Shuler et al. [73], also concluded that IM-VHEET generally outperforms VMAT. Palma et al. compared a 16-beam, 100 MeV IM-VHEET plan to VMAT, achieving similar PTV coverage and dose homogeneity. Shuler et al. reported a mean integral dose reduction of 14% and 16% compared to VMAT for 100 and 200 MeV beams, respectively. The 100 MeV IM-VHEET plan showed mean dose reductions to the esophagus, right lung, and left lung by 3, 20, and 12%, respectively, while the 200 MeV plan resulted in reductions of 7%, 21%, and 16% for the same OARs. Additionally, relative to VMAT, Spek et al. [171] found that a nine-beam, 400 MeV IM-VHEET plan reduced doses to the esophagus and spinal cord but increased the dose to the plexus.

Bohlen et al. [177] reported on IM-VHEET plans with 3, 5, 7, or 16 VHEE beams at energies of 100 or 200 MeV. The chosen configurations were 5 × 100, 3 × 200, 5×200, 7 × 200, and 16 × 200 MeV. Compared to photons, the mean dose to the body, lung, and heart was higher in the 5 × 100 IM-VHEET plan, while the mean dose to the esophagus and trachea was higher in the 3 × 200 plans. All other dose metrics were almost unaltered between the two modalities, with only the 16 × 200 IM-VHEET plan yielding a 2.1% improvement in PTV coverage. In the context of VHEET lung studies, the findings of Bohlen et al. appear slightly less encouraging. This study, however, compared the 3D-CRT modality for VHEET with the standard-of-care IMRT for photons. As a result, the plan quality of 3D-CRT VHEET plans is substantially degraded relative to IMRT, which is typically the case when comparing photon 3D-CRT and IMRT treatments. Similarly, IM-VHEET plans are superior to 3D-CRT VHEET plans. The 3D-CRT modality for both VHEE and photon beams was also considered by Moskvin et al. [173], who reported a 7.1% VHEET-averaged improvement in normal tissue sparing.

Shuler et al. compared IM-VHEET plans, at energies of 100 and 200 MeV to IMPT, finding the mean integral dose for IM-VHEET to be steadily about 50% higher. For the OARs, the 200 MeV IM-VHEET plan resulted in a 5% mean dose to the left lung, while the bronchial tree and esophagus received 5% and 8% higher doses, respectively.

**Table 5 cancers-17-00181-t005:** Clinical studies applying VHEET treatments for lung cancer.

Study	VHEE Beam	VHEE Beam	VHEET	Others RT	FLASH?	Dose
	Energy [MeV]	Number	Technique	Modalities		Engine
Moskvin et al. [173]	200	8	3D-CRT VHEET	6 MV 3D-CRT	NO	PENELOPE
Bazalova-Carter et al. [174]	100	36	IM-VHEET	6 MV VMAT	YES	EGSnrc + RayStation ^1^
Palma et al. [72]	100	16	IM-VHEET	10 MV VMAT	YES	EGSnrc + RayStation ^1^
Schuler et al. [73]	100, 200	16	IM-VHEET	(a) 6 MV VMAT	YES	EGSnrc + RayStation ^1^
				(b) IMPT		
Bohlen et al. [177]	(1) 100	(1) 5	3D-CRT VHEET	6 MV VMAT	YES	eMC RayStation ^1^
	(2) 200	(2) 3, 5, 7, 16				
Zhang et al. [176]	100, 120, 140	11	IM-VHEET	NO	YES	Geant4
Spek et al. [171]	100, 200, 300, 400	9, 18, 36	IM-VHEET	VMAT	NO	FLUKA

Studies listed in chronological order. ^1^ Research version.

### 9.4. Pediatric Brain Tumors

Bazalova-Carter et al. [174] and Schuler et al. [73] evaluated two pediatric brain tumor cases, that is, a glioblastoma and a posterior fossa ependymoma, respectively. Bazalova-Carter et al. compared IM-VHEET plans comprising 13 VHEE beams at energies of 60, 80, 100, or 120 MeV, as well as IM-VHEET plans with 13, 17, or 36 VHEE beams at 80 MeV. The best configuration was the 36×100 MeV. Compared to VMAT, the IM-VHEET plan was more conformal around the target, with a 33% integral dose reduction. Shuler et al. considered IM-VHEET plans with 16 beams at energies of 100 or 200 MeV and found a similar integral dose between the VMAT and IM-VHEET modalities. Regarding the OARs, Bazalova-Carter et al. found that IM-VHEET reduced the mean doses to the cochleae and temporal lobes by 70 and 33%, respectively. According to Shuler et al., for a beam energy of 100 MeV, the brain, brainstem, and spinal cord showed a modest mean dose decrease compared to VMAT. Conversely, the chiasm and left and right parotid glands had mean dose reductions of 75, 5, and 36%, respectively. Higher reductions were reported for 200 MeV VHEE beams.

Clements et al. [178] simulated the treatment plan of a pediatric glioblastoma using a GRID-fractionated technique with sub-millimeter-sized VHEE beamlets (minibeams). A cylindrical tungsten collimator, comprising a 2D rectangular array of parallel holes, was placed along the VHEE beamline close to the patient’s surface. A second, tumor-shaped, cutout collimator adjusted the contour of the minibeam pattern as they exited the cylinder. VHEE beams, 150 and 200 MeV in energy, were used both in the minibeam and open beam (IM-VHEET) configurations and were compared to VMAT. Results showed that the averaged OAR doses for the 150 and 250 MeV VHEE beams, respectively, were 17 and 38% higher in the GRID setup than in VMAT. However, with respect to VMAT, the open beam configuration led to an average OAR dose that was 25 and 22% lower at 150 and 200 MeV, respectively. Overall, unlike open-beam VHEE plans, mini-GRID VHEET performed worse than VMAT. Nevertheless, from a broader perspective, minibeam spatially fractionated RT (SFRT) may offer advantages (e.g., lower toxicity, enhanced tumor control, improved immune response, and hypoxia mitigation), supporting further studies.

Compared to IMPT, Shuler et al. reported a 29 and 39% mean dose reduction to the spinal cord, and a 16 and 19% mean dose reduction to the chiasm, for the 100 and 200 MeV VHEE beams, respectively. However, IMPT outperformed IM-VHEET in terms of dose metrics for the brain, cochlea, and parotid glands.

**Table 6 cancers-17-00181-t006:** Clinical studies applying VHEET treatments for pediatric brain tumors, non-pediatric intracranial tumors, and cancers of the head and neck, liver, esophagus, and anus.

Study	VHEE Beam	VHEE Beam	VHEET	Others RT	FLASH?	Dose
	Energy [MeV]	Number	Technique	Modalities		Engine
	Pediatric Brain					
Bazalova-Carter et al. [174]	(1) 60, 80, 100, 120	(1) 13	IM-VHEET	6 MV VMAT	YES	EGSnrc + RayStation ^1^
	(2) 80	(2) 13, 17, 36	IM-VHEET	6 MV VMAT	YES	EGSnrc + RayStation ^1^
Schuler et al. [73]	100, 200	16	IM-VHEET	(a) 6 MV VMAT	YES	EGSnrc + RayStation ^1^
				(b) IMPT		
Clements et al. [178]	150, 200	12	(a) SFRT (GRID)	6 MV VMAT	YES	TOPAS
			(b) IM-VHEET			
	Non-pediatric intracranial tumors					
Palma et al. [72]	120	32	IM-VHEET	(a) 6 MV VMAT	YES	EGSnrc + RayStation ^1^
				(b) 6 MV CyberKnife		
Bohlen et al. [20]	(1) 100	(1) 5	3D-CRT VHEET	3 MV VMAT	YES	eMC RayStation ^1^
	(2) 200	(2) 3, 5, 7, 16				
Muscato et al. [157]	(1) 110-100 ^2^	(1) 3	IM-VHEET	(a) 6 MV IMRT	YES	FLUKA
	(2) 90–100–110 ^2^	(2) 7		(b) IMPT	YES	
	(3) 90–120 ^2^	(3) 4				
	(4) 60–80–220 ^2^	(4) 7				
Krim et al. [123]	200	2	3D-CRT VHEET	NO	YES	GATE
Bohlen et al. [25]	50, 100, 150, 200, 250	7	TR-VHEET ^2^	TR-PT ^2^	YES	RayStation ^1^
	Head and Neck					
Zhang et al. [176]	80, 100, 120	16	IM-VHEET	NO	YES	Geant4
	Anus					
Palma et al. [72]	100	32	IM-VHEET	15 MV VMAT	YES	EGSnrc + RayStation ^1^
	Esophagus					
Palma et al. [72]	120	32	IM-VHEET	6 MV VMAT	YES	EGSnrc + RayStation ^1^
Bohlen et al. [25]	50, 100, 150, 200, 250	7	TR-VHEET ^3^	250 MeV TR-PT ^3^	YES	RayStation ^1^
	Liver					
Palma et al. [72]	120	32	IM-VHEET	10 MV VMAT	YES	EGSnrc + RayStation ^1^

Studies listed in chronological order (within each section). ^1^ Research version. ^2^ Energy modulation. ^3^ Transmission planning (TR): pencil beam-scanning without intensity modulation.

### 9.5. Non-Pediatric Intracranial and Head Tumors

Palma et al. [72] considered an acoustic neuroma and compared the following plans: IM-VHEET with 32 beams at 120 MeV, CyberKnife, and VMAT. The conformity indices of IM-VHEET were higher than those of the CyberKnife but lower than those of VMAT. PTV coverage was nearly equal across all plans. The VHEE mean dose to the brainstem was the lowest, while the VHEE mean dose to the right cochlea was 15% higher than that of VMAT but 12% lower than that of CyberKnife. For a glioblastoma case, Bohlen et al. [177] compared 3D-CRT IM-VHEET to IMRT. For 3D-CRT, 3, 5, 7, and 16 VHEE beams at 100 or 200 MeV were selected. In the 5 × 100 MeV 3D-CRT VHEET plan, the mean doses to the body and the OARs were higher compared to those of IMRT. However, by increasing the VHEE beam energy at 200 MeV, all dose metrics significantly improved. For example, in the 5 × 200 MeV 3D-CRT VHEET plan, the mean doses to the brainstem, chiasm, and optic nerve were 46.1, 46.7, and 34.8% lower than those of IMRT, respectively. For the 16 × 200 set-up, the same reductions were 52.5, 42.9, and 27.2%. In a subsequent paper, Böhlen et al. [25] compared proton and VHEET plans for a glioblastoma case using the transmission modality (see Section 9.7.3). VHEET plans ranged from 50 to 250 MeV in 50 MeV steps, with seven equispaced co-planar beams. Glioblastoma represents a shallow target, and as such, acceptable conformity can be achieved using VHEE beams of 100 MeV and above. For VHEET energies of 100–250 MeV, only small dose metric differences were reported between the two modalities.

Muscato et al. [157] evaluated IM-VHEET, IMRT, and IMPT for targeting meningioma and skull base chordoma. The chordoma is a particularly complex case, where OAR-sparing requirements limit PTV coverage. Priority was given to minimizing the absorbed dose to the brainstem and spinal cord, even if the desired PTV coverage was not achieved. IM-VHEET plans had three beams (100 and 110 MeV) or seven beams (90, 100, and 110 MeV) for the meningioma, and four beams (90 and 120 MeV) or seven beams (60, 80, and 120 MeV) for the chordoma. For the meningioma case, PTV coverage and OAR dose metrics were of comparable quality among all modalities, with IM-VHEET slightly underperforming in PTV coverage. Within IM-VHEET, the PTV coverage of the four-beam plan (98.97%) was slightly superior to that of the seven-beam plan (97%). For the chordoma case, the IM-VHEET plan provided the lowest PTV coverage: 85 and 87% for the four-beam and seven-beam IM-VHEET plans, respectively, compared to 92.96 and 93.57% for IMRT and IMPT, respectively. The four-beam IM-VHEET plan resulted in low OAR dose metrics, comparable to those of IMPT. Conversely, the seven-beam IM-VHEET plan yielded high OAR dose metrics, similar to those of IMRT. Overall, even with a small number of beams and beam energies lower than 130 MeV, IM-VHEET plans could approach the quality of IMRT/IMPT. Krim et al. [123] evaluated a two-beam 3D-CRT VHEET plan for a virtual brain tumor. VHEET was considered with both collimated and focused beams. Results showed that focused VHEE beams could precisely concentrate high-dose peaks into small volumetric targets while sparing adjacent healthy tissues. Compared to collimated VHEE beams, focusing significantly reduces both the entrance and exit doses.

For a head case, Zhang et al. [176] evaluated the dose metrics of a 16-beam IM-VHEET plan at energies of 80, 100, or 120 MeV. All plans satisfied the clinical requirements, with similar OARs and external volumes dose metrics. The 16 × 80 MeV was associated with the best PTV dose distribution.

### 9.6. Gastrointestinal Tumors

Palma et al. [72] considered IM-VHEET in the anus, esophagus, and liver cancer. The liver case had a single PTV, while the anal and esophagus case had two PTVs. Compared to VMAT, the following IM-VHEET plans were evaluated: anus, 32 beams at 100 MeV, and esophagus and liver, 32 beams at 120 MeV. In both anal and esophagus sites, the IM-VHEET plan excelled in conformity and sparing of external volumes. For example, in the anal case, IM-VHEET reduced the mean doses to the genitalia and perineum by 21.8 and 18.5%, respectively. Similarly, in the esophagus case, the mean doses to the spinal cord and heart were reduced by 23 and 20%, respectively. In the lung, the PTV dose was superior with IM-VHEET, and the OAR mean doses were very similar, with the most significant difference being a 19% higher chest wall dose for VMAT. In the anal and esophagus cases, IM-VHEET also reduced the integral dose by 13 and 22%, respectively. In the liver cancer case, however, no significant variation was reported. Böhlen et al. [25] also examined an esophagus case, comparing VHEET plans with energies of 50, 100, 150, 200, and 250 MeV to a proton plan. All plans utilized seven beams and were designed according to the transmission modality. VHEET energies between 100 and 250 MeV are needed for dose metrics to be comparable to those of proton therapy, while a VHEET energy of 250 MeV is required to achieve similar conformity.

### 9.7. VHEE FLASH Radiotherapy in the Clinic

#### 9.7.1. IM-VHEET Treatment Time

The studies by Bazalova-Carter et al. [174], Palma et al. [72], and Shuler et al. [73] evaluated the IM-VHEET treatment time, finding it potentially shorter than that of photon-based treatments. Assuming a dose–rate of 14 Gy/min for the Varian TrueBeam linac in the 6 MV FFF mode with a 6% X-ray beam production efficiency, Bazalova-Carter et al. estimated the dose–rate of a 100 MeV VHEE beam to be approximately 117 Gy/s. Therefore, if scanning time is ignored, large VHEE treatment doses could be delivered in sub-second times, which is theoretically compatible with FLASH applications.

#### 9.7.2. 3D-CRT VHEET Treatment Modality

Like IMRT, IM-VHEET may require an extended delivery time, potentially reducing or eliminating the FLASH effect. Ronga et al. [9] clarified that to trigger the FLASH effect within a 1 L volume, a single beam comprising approximately 2500 VHEE pencil beams with 2 mm spacing would need to be delivered within 100 ms. Using a step-and-shoot approach would require a scanning speed of at least 5.1 m/s without pauses, implying a minimum repetition rate of 25 kHz. Since no current system can meet such requirements, only small sub-volumes could receive UHDR irradiation in this setup. DesRosiers et al. [172], Moskin et al. [173], Garnica−Garza [170], and more recently, Bohlen et al. [177], presented VHEET treatments in the 3D-CRT modality. Garnica−Garza found the non-tumoral integral dose (NTID) to be lower in 3D-CRT VHEET than in 3D-CRT photons. As the photon NTID is independent of the RT modality [179], the authors reasonably postulated that VHEET NTID is superior to both photon 3D-CRT and IMRT plans. In brain cancer, Bohlen et al. showed that even a 3D-CRT VHEET treatment with 3 to 7 beams may provide dose distributions competitive with IMRT. In lung cancer, the same authors estimated the differences in OAR dose metrics, between VHEE 3D-CRT and IMRT, mostly below 10%, which is small compared to the potential FLASH-sparing effect. Thus, the increased biological selectivity of FLASH, due to normal tissue sparing, could compensate for or even surpass the dose metrics of IMRT. Moreover, the FLASH effect is prominent in the high dose region, and a biologically selective sparing of healthy tissues in the GTV-to-PTV margin could be achieved, even if 3D-CRT plans fail to provide the steep PTV dose gradients of IM-VHEET.

#### 9.7.3. Transmission VHEET Treatment Modality

Transmission proton therapy (TR-PT), also known as transmission FLASH [180], is a proton planning modality where the patient’s tissue is irradiated with the beam section proximal to the Bragg peak, and the Bragg peak is preferably kept outside the patient. TR-PT sacrifices IMPT conformity to allow UHDR delivery [181]. Böhlen et al. [25] were the first to compare the TR modality in VHEET and proton plans. The authors demonstrate that transmission VHEET (TR-VHEET) plans with energies in the 150–250 MeV range may provide acceptable dosimetric plan quality compared to TR-PT plans.

**Table 7 cancers-17-00181-t007:** Clinical VHEET studies with implications for FLASH therapy and/or directly considering the FLASH effect.

Study	Treatment Site	FLASH-Specific Actions/Considerations
	Non-FLASH studies with FLASH implications	
Bazalova-Carter et al. [174]	Lung, prostate, pediatric brain	UHDR esteemed (dose–rate ∼117 Gy/s)
Palma et al. [72]	Acoustic neuroma, liver, esophagus, anus	UHDR esteemed (dose–rate ∼117 Gy/s)
Schuler et al. [73]	Prostate, lung, head and neck, pediatric brain	UHDR esteemed (dose–rate ∼117 Gy/s).
DesRosiers et al. [172]	Prostate	3D-CRT treatments: easy CONV-to-FLASH transition
Moskvin et al. [173]	Prostate, lung	3D-CRT treatments: easy CONV-to-FLASH transition
Garnica-Garza [170]	Prostate	3D-CRT treatments: easy CONV-to-FLASH transition
Bohlen et al. [177]	Lung and brain	3D-CRT treatments: easy CONV-to-FLASH transition
Bohlen et al. [25]	Prostate, brain, esophagus	TR ^1^: easy CONV-to-FLASH transition
	FLASH studies	
Sarti et al. [175]	Prostate	DMF application ^2^
Muscato et al. [157]	Non-pediatric intracranial tumors	FMF application ^3^
Clements et al. [178]	Pediatric brain	UHDR (>40 Gy/s) with GRID-RT
Zhang et al. [176]	Lung, head	UHDR (>40 Gy/s)

Studies listed in chronological order (within each section). ^1^ Transmission planning (TR): pencil beam-scanning without intensity modulation. ^2^ Dose-modifying factor (DMF). ^3^ FLASH-modifying factor (FMF).

#### 9.7.4. VHEET with Dose/FLASH-Modifying Factors

The Dose-Modifying Factor (DMF) [182] is defined as the ratio of doses required at UHDR and CONV rate to achieve an isoeffect for a given biological system. The inverse of the DMF, known as the FLASH-modifying factor (FMF) was first introduced by Bohlen et al. [183] and is analogous to the RBE definition for different radiation qualities. A DMF, or FMF, of 1 corresponds to the CONV regime. Sarti et al. [175] and Muscato et al. [157] evaluated UHDR in IM-VHEET by employing a set of DMFs, with values computed according to the most recent finding on healthy tissues sparing under FLASH conditions. Sarti et al. optimized IM-VHEET treatment plans with DMFs of 0.6, 0.7, 0.8, 0.9, and 1, while Muscato et al. used a DMF value of 0.8 (corresponding to an FMF of 1.25). In both studies, the IM-VHEET plan at CONV dose rates was modified as follows: in each voxel outside the PTV, the absorbed dose was multiplied by the DMF, applying the same factor to all tissues. Sarti et al. found that with a DMF of 0.8, even an 80 MeV IM-VHEET plan could, in theory, compete with IMRT. Muscato et al. considered two clinical situations, leading to different treatment approaches. In the first case, the CONV IM-VHEET plan already provided satisfactory PTV coverage, and the FLASH effect further reduced OAR dose metrics, increasing OAR sparing. In the second case, the CONV IM-VHEET plan had limited PTV coverage due to OAR dose limits, and the FLASH effect allowed the overall dose to be rescaled, improving PTV coverage until OAR dose metric limits were reached.

#### 9.7.5. GRID-Based VHEET at FLASH UHDR

Using TOPAS MC simulations, Clements et al. [178] modeled UHDR in the PTV for both multi-GRID and open beam VHEE configurations, with a 0.75 μA beam current. Regarding the OARs, the open beam set-up had UHDR higher than 40 Gy/s in 8/9 and 9/9 OARs, for 150 and 250 MeV VHEE beams, respectively. In multi-GRID modality, the peak (in-beam) dose–rates were likely above 40 Gy/s. However, because an OAR volume-averaged dose–rate encompasses all OAR structures, including low-dose valleys, in the multi-GRID setup only 3/9 OARs had volume-averaged dose–rates higher than 40 Gy/s.

#### 9.7.6. VHEET Total Treatment Time at FLASH UHDR

Using the Geant4 toolkit, Zhang et al. [176]) performed a quantitative comparison of different VHEE beam energies concerning FLASH-related parameters, specifically the dose averaged dose–rate (DADR) and the total treatment time (TTT). For a head IM-VHEET treatment with 4655 scanning spots and beam energy values of 80, 100, and 120 MeV, a minimum beam intensity of ∼2.5 × $10^{11}$ electrons/s is needed for more than 90% of the PTV volume to be irradiated under FLASH conditions (DADR > 40 Gy/s). This corresponds to TTT values of 5258.75, 5149.75, and 4976.75 ms (including a scanning “repositioning” time of 872.75 ms), for the three selected VHEE energies, respectively. For a lung treatment with 1591 scanning spots and beam energies of 100, 120, and 140 MeV, the minimum beam intensity was ∼9.37 × $10^{11}$ electrons/s, and the TTT values were 1034.25, 981.55, and 928.15 ms (including a scanning time of 298.75 ms), for the three energy values, respectively. Results were reported for a 10 Gy fraction dose and a constant beam intensity per spot. In both PTV and OARs, a higher VHEE beam energy leads to a higher dose–rate, thus shortening the beam-on delivery time.

## 10. Conclusions

VHEE beams, due to their advantageous dosimetric properties, are considered innovative RT modalities. Early concerns about secondary radiation, such as neutron contamination and induced radioactivity, have been mitigated by experimental and simulation studies, which show doses are negligible, often lower than natural background levels. Similarly, secondary radiation along the VHEE beamline is minimal, with ambient dose levels in VHEET rooms comparable to those in proton therapy. A key challenge of VHEET, particularly with single-beam setups, is the high surface dose. This can be addressed by incorporating beam-focusing systems, i.e., magnetic quadrupoles, which reduce the entrance dose, minimize lateral scattering, and allow precise targeting of small 3D volumes.

The development of compact clinical VHEE systems aims to make VHEET both accessible and affordable. Two main acceleration concepts are being explored: RF-based linear accelerators and laser-driven LWFA plasma accelerators. RF linacs are established technologies offering high beam quality and stability, with advanced C/X-band systems showing promise for compact clinical applications. LWFA systems, on the other hand, sustain much higher accelerating fields, offering potential benefits in compactness and cost. However, they face challenges related to beam stability and a broad energy spectrum.

Currently, no medically certified TPS exists for VHEET, with most studies relying on MC simulations. The clinical implementation also requires accurate secondary dosimeters. While radiochromic film shows energy independence at VHEE levels, their clinical use is limited by lengthy post-irradiation processing. Conversely, ionization chambers, which provide immediate feedback, suffer from high recombination losses. The search for an optimal clinical dosimeter is ongoing, with promising prototypes under development.

To date, VHEET planning evaluations have primarily focused on dose distribution in complex anatomies, comparing VHEET with standard-of-care photon RT and, to a lesser extent, proton therapy. The general findings from published studies, representing overall trends rather than specifics, are as follows:Compared to photon therapy, VHEET offers a more conformal target dose distribution and comparable or improved target coverage, often with better sparing of OARs. However, VHEET provides less conformal dose distribution and inferior/comparable target coverage than proton therapy. These results apply when particle plans within the same treatment modality are evaluated, such as 3D-CRT or IMRT.Increasing the number of beams improves PTV coverage but may raise OAR doses, which can be reduced by increasing beam energy. VHEET plans generally use the same beam arrangements as photon plans, though further research is needed to develop an optimal VHEET strategy that may differ slightly from photon therapy approaches.Higher VHEE energy levels, near the 250 MeV limit or beyond, are often linked to better plan quality and improved OAR sparing. If confirmed, this could have significant technological and cost implications.

The FLASH effect, globally regarded as one of the most promising advancements in RT, is expected to find in VHEET its ideal application. Designing VHEET systems capable of triggering the FLASH effect can leverage the knowledge gained from low-energy electron FLASH systems. However, the different energy ranges of VHEE beams pose significant additional challenges. Compared to CONV VHEET systems, FLASH VHEET systems require precise tuning of time-dependent dose parameters, introducing new technological hurdles. Therefore, the transition from low-energy electron FLASH to VHEE FLASH, as well as from CONV VHEET to FLASH VHEET, is not going to be straightforward. Establishing a multidisciplinary FLASH-centered roadmap for VHEET development is essential to address all involved complexities.

***Radiobiology***. Studies are required to assess the FLASH effect in VHEET, specifically examining how treatment temporal parameters influence its occurrence.***Clinical***. A therapeutic protocol is essential to guide clinicians in selecting the appropriate irradiation geometry for FLASH RT. A key consideration is whether single-beam arrangements will be the primary approach for FLASH or if multi-beam plans should also be considered. Additionally, it must be determined whether plan quality should be compromised for shorter delivery times and if intensity-modulated techniques should be replaced by 3D-CRT or transmission techniques.***Technological***. Efforts should focus on creating stable, compact, and cost-efficient VHEE systems that are capable of sustaining UHDRs.***Clinical/technological***. The development of a medically certified VHEE TPS that accounts for the FLASH effect is crucial, with healthy tissue doses adjusted to reflect the sparing effect at UHDR.***Technological***. Compact, easy-to-use, online beam monitoring systems, energy-independent in the VHEE energy range and capable of sustaining UHDR regimens, need to be developed.

The economic impact of new therapies is crucial. The charge-to-mass ratio of electrons, which is 1,836 times higher than that of protons, results in 3–4 times lower magnetic rigidity, facilitating particle acceleration and steering. This enables the development of smaller, more cost-effective VHEE systems for both CONV and UHDR treatments. This advantage is particularly significant for FLASH therapy, as it allows for multi-directional beams at UHDR within the required time limits.

In the years to come, photon and proton RT technologies will continue to advance, but improvements in RF linacs and laser-driven systems suggest even faster development of VHEET technologies, driven by the growing demand for VHEE systems in various fields of physics. This technological progress, combined with the unique dosimetric potential of VHEE beams, inspires confidence in the future tuning and integration of VHEET in the existing clinical environments.

## Figures and Tables

**Figure 1 cancers-17-00181-f001:**
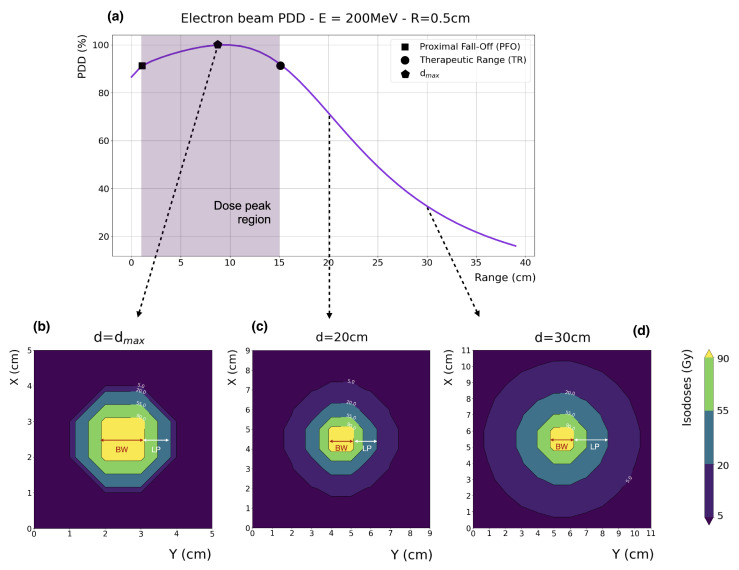
(**a**) Longitudinal *on-axis* dose distribution of a collimated VHEE parallel beam impinging on a water phantom (with vacuum beforehand); the beam has an energy of 200 MeV and a 1 cm diameter, showing the $d_{max}$ (maximum dose depth), PFO (proximal fall-off at 90% of the dose at $d_{max}$), and TR (therapeutic range at 90% of the dose at $d_{max}$). The transversal dose distribution is shown at depths of (**b**) $d_{max}$, (**c**) 20, and (**d**) 30 cm. The LP (lateral penumbra) is the distance between the 90% and 20% intensity levels, whereas the BW (beam width) is the width at 90% of the maximum dose value at that depth.

**Figure 3 cancers-17-00181-f003:**
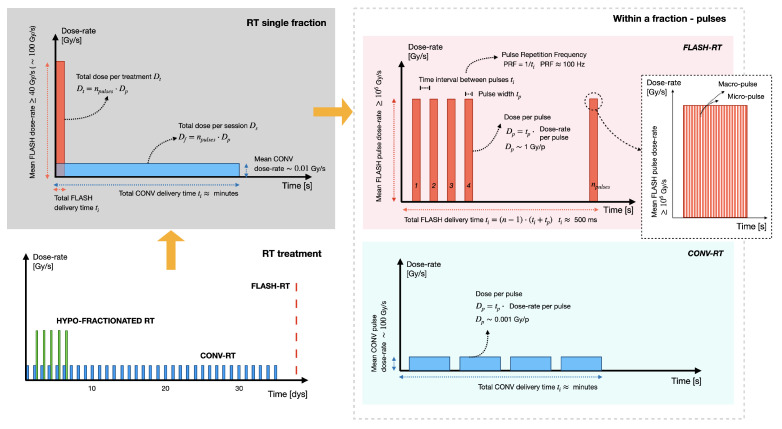
FLASH vs. CONV-RT modality: inter-dependent temporal parameters characterizing the entire RT treatment, a single fraction, and a single pulse.

**Figure 4 cancers-17-00181-f004:**
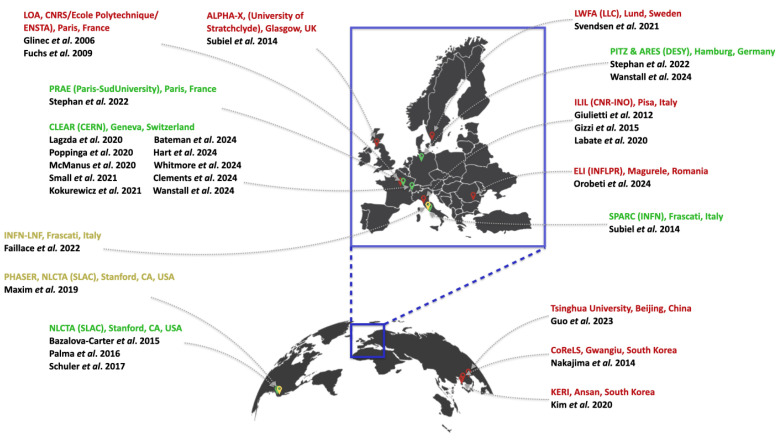
Facilities currently being used worldwide for VHEE beam production are categorized as follows: *Green* for RF-based research linear accelerators (RF linacs) [12,18,19,24,31,62,64,66,67,68,69,70,71,72,73,74], *yellow* for compact FLASH VHEET-dedicated RF linac platforms [10,11], and *red* for research laser–plasma accelerators exploiting the Wakefield effect [13,15,16,21,31,63,75,76,77,78,79]. Listed under each facility are the papers based on research conducted at that center.

**Figure 5 cancers-17-00181-f005:**
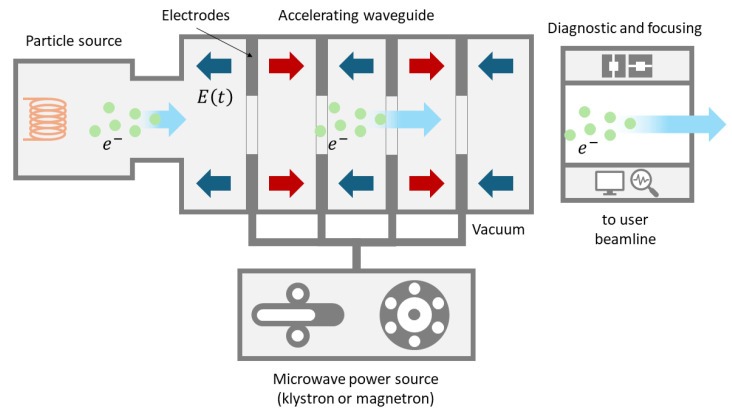
Schematic representation of a linear accelerator (*linac*) and its main components: Electrons are generated by a particle source within a vacuum chamber, using either a cold cathode, a hot cathode, a photocathode, or an RF source. A microwave source, such as a klystron or magnetron, powers the accelerating waveguide, a series of open-ended cylindrical electrodes guiding the electron beam toward the diagnostic and manipulation (focusing) area. The arrows represent the electric field oscillating (red/blue) direction between the electrodes of the linac powered by the RF generator. Finally, the VHEE beam reaches the user beamline.

**Figure 6 cancers-17-00181-f006:**
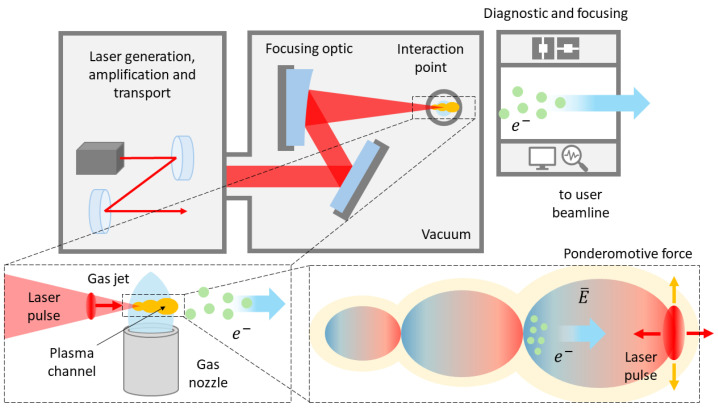
Schematic representation of a laser Wakefield accelerator with its main components. An ultra-short (∼10–40 fs) laser pulse is generated, amplified, and transported to the interaction area. The laser pulse is then focused on a gas target, usually by an off-axis parabolic mirror. Electrons are accelerated within the plasma and propagate in the forward direction.

**Figure 7 cancers-17-00181-f007:**
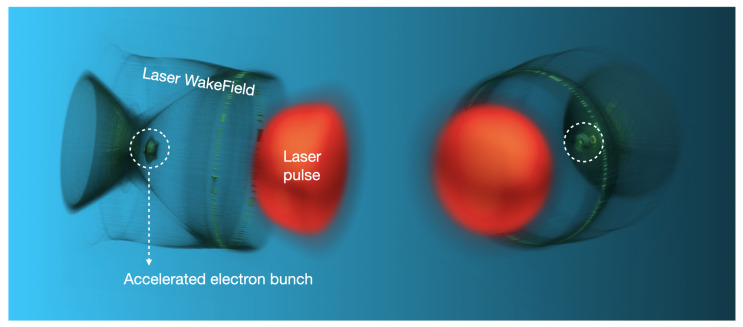
Particle-in-cell simulation [102] artwork of laser Wakefield electron acceleration. The laser pulse propagates in an underdense “transparent” plasma, generating a strong longitudinal electric field, whose velocity is close to *c*, the *Wakefield*. When electrons are trapped within the accelerating region of the electric field, their energy is boosted to very high values over very short distances.

**Figure 8 cancers-17-00181-f008:**
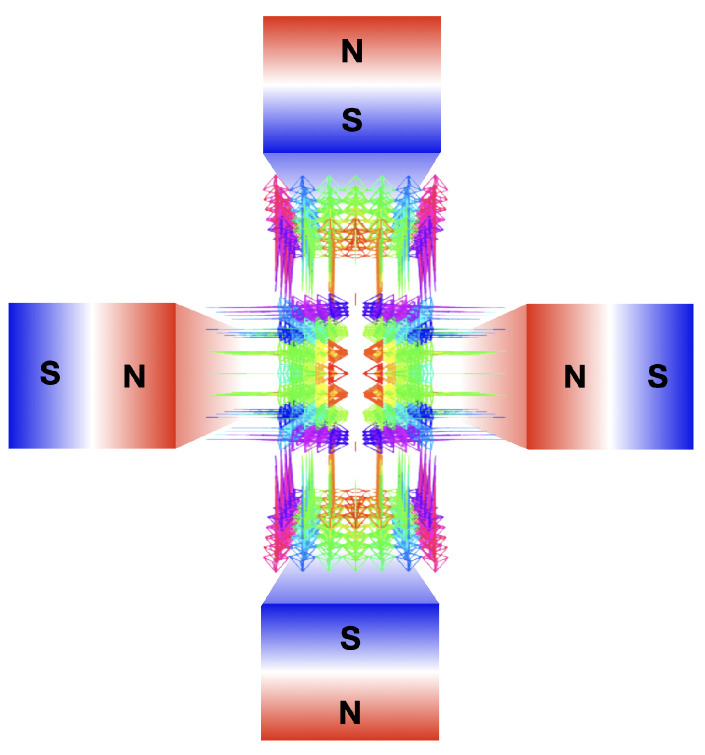
Quadruple magnetic field lines. Colors represent the magnetic field direction and strength.

**Figure 9 cancers-17-00181-f009:**
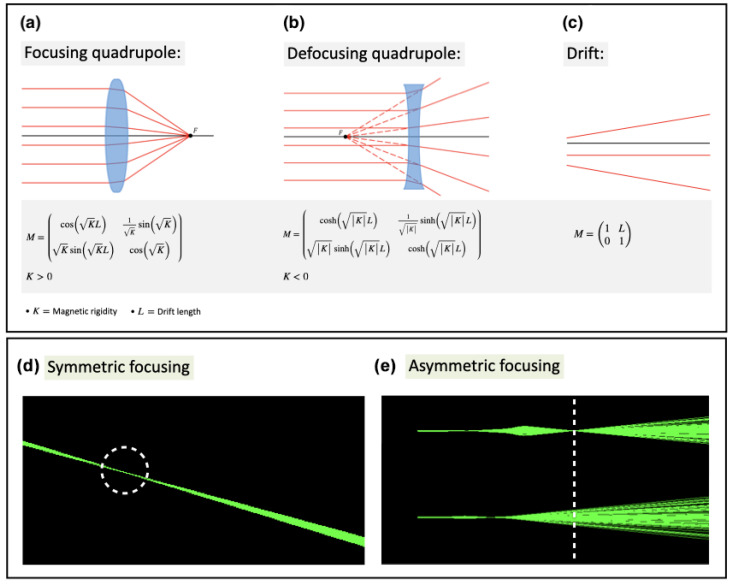
VHEE beam focusing: (**a**) Focusing and (**b**) defocusing magnetic quadrupoles, graphic rendering from the thin lens approximation, and matrix formalism. (**c**) Drift graphic rendering and matrix formalism. The final focusing can be symmetric (**d**), meaning the beam converges along both planes or asymmetrically (**e**), with the beam converging along one plane only.

**Table 1 cancers-17-00181-t001:** Types of VHEET beams.

VHEE Beam Type:	Description:
**Divergent**	The beam widens transversely as it travels before reaching the patient.
**Non-divergent/collimated/parallel**	The beam maintains its transverse size constant before reaching the patient.
***Asymmetrically*****/*****Line*** **focused**	The beam converges along one axis and either diverges or remains collimated along the other.
***Symmetrically*** **focused**	The beam converges along both axes.

**Table 2 cancers-17-00181-t002:** RF linac-based VHEE facilities: beam parameters and RT studies.

Site	Energy	Energy	Pulse	Pulse	Repetition	Bunches ×	Bunch	Bunch Length	Bunch	Normalized	VHEE RT
		**Spread**	**Charge**	**Length**	**Rate**	**Pulse**	**Charge**	**(rms)**	**Frequency**	**Emittance**	**Studies**
CLEAR [82]	60 to 220 MeV	<0.2%	up to 75 nC	0.1 ps to 100 ns	0.8333 to 10 Hz	1 to ∼150	0.005 to 1.5 nC	0.1 to 10 ps	1.5 or 3 GHz	1–20 μm	Lagzda et al. [24], Poppinga et al. [66],
											Mcmanus et al. [67], Small et al. [68],
											Kokurewicz et al. [69], Bateman et al. [64],
											Hart et al. [18], Rieker et al. [97], Whitmore et al. [12],
											Clements et al. [70], Corsini et al. [98]
NLCTA [83]	50 to 120 MeV [85]	0.6 to 1%	100 pC [99]	0.125 μs	10 Hz	1440		26 fs [85]	0.714 GHz	1 ${\mathrm{mm}}_{\cdot }$mrad [85]	Bazalova-Carter et al. [71], Palma et al. [72],
											Schuler et al. [73]
PITZ [88]	22 MeV ^1^	1 to 2 keV [89]	0.1 to 5000 pC	up 1 ms	1 to 10 Hz	1 to 1000		<1 to ∼30 ps	100 kHz to 1 MHZ	∼0.7 ${\mathrm{mm}}_{\cdot }$mrad	Stephan et al. [74]
ARES [91]	45 to 155 MeV	<0.2% [92]	0.003 to 200 pC		1 to 50 Hz			30 fs		0.1–1 μm [90]	Wanstall et al. [19]
SPARC [94]	5 to 180 MeV	0.1–5 MeV			10 Hz	1 to 5	10 pC to 2 nC	20 fs to 10 ps	50 MHz to 2 THz	0.3–10 μm	Subiel et al. [31]
PRAE [96]	50 to 70 MeV ^2^	<0.2%	0.05 pC to 2 nC	<10 ps		1				3–10 ${\mathrm{mm}}_{\cdot }$mrad	Han et al. [100]

Where a value in the table is missing a reference, it corresponds to the one associated with the facility name. ^1^ Planned upgrade to 250 MeV. ^2^ Planned upgrade to 140 MeV.

**Table 3 cancers-17-00181-t003:** Laser-based VHEE facilities: beam parameters and RT studies.

Facility	Rough Average	Energy	Max Laser Pulse	Laser Pulse	VHEET RT
	Energy *	Spread	Duration **	Repetition Rate ***	Studies
LOA Salle jaune [115]	(1) 170 MeV	(1) 40 MeV (FWHM)	30 fs	10 Hz	(1) Glinec et al. [13]
	(2) 170 MeV	(2) 15% (FWHM)			(2) Semushin and Malka [116], Fuchs et al. [21]
	(3) 120 MeV	(*3*) 20 MeV (FWHM)			(3) Lundh et al. [105]
ALPHA-X [117]	135 MeV	44 MeV (FWHM)	35 fs	0.33 Hz	Subiel et al. [31]
KERI [119]	94 MeV	80 MeV (σ)	40 fs	10 Hz	Kim et al. [78]
ILIL [120]	100 MeV	50–250 MeV energy distribution	30 fs	2.5 Hz	Labate et al. [16]
Lund High-Power Laser Facility [121]	95 MeV	50 to 150 MeV energy distribution	37 fs	10 Hz	Svendsen et al. [15]
Tsinghua University [77]	75 MeV	0 to 175 energy distribution	25 fs	10 Hz	Guo et al. [77]

* These values were estimated based on the published energy plots. ** The duration of the laser pulse used for acceleration is not representative of the bunch duration but is provided as an indication of the acceleration regime. *** An upgrade to 100 Hz is potentially foreseen for all these facilities.

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
