# Peer review of "Very High-Energy Electron Therapy Toward Clinical Implementation"

_cancers, 2025, doi:10.3390/cancers17020181_

Round 1
Reviewer 1 Report
Comments and Suggestions for Authors
Summary
This is a white paper on a wide topic of VHEE (Very High Energy Electron beams). It presents a very comprehensive information on all details of this topic. Therefore, it is very extensive (28 pages of pure text plus figures and extensive list citations, together 46 pages) and comes into deep details of all issues of the topic.
The extent of the article strongly compromises lucidity of the information transferred to the reader.
All statements presented in this survey are well supported by the literature. The review provides detailed information for various specialties – physicists, preclinical researchers, clinicians – researchers. Any of these specialists may not capture the entire detailed information and may not find it useful. Thus, the extent and comprehensiveness of the article becomes very questionable.
This white paper is more relevant as a chapter in a textbook than an article in oncology journal.
General concept
There is a reasonable concept of the survey. It describes physical properties, technical issues, dosimetry, biology of the effect, preclinical studies, achieved results and future perspectives of very high energy electron beams in a logical order. A huge body of literature is referred. This white paper may serve as a good source of information.
Review, specific comments
This survey is very comprehensive and detailed (as presented in the summary). It comes into all deep details of VHEE. The title “……. a review study” is a bit irrelevant, there is no question stated and answered as would be relevant in a “study”. It is indeed nothing else than a review.
Each specialist (physicist, biology researcher, radiation oncologist) may find a very detailed information relevant to own specialty, however the remaining information may find abundant. It is questionable how far is this format suitable for the cancer journal.
The review comes into details that may not be necessarily discussed. E.g.:
- Description of flash effect and FLASH-RT (rows 262-285). It may be enough to refer to relevant publications. (A reader able to understand this survey definitely knows what means FLASH.)
- The listing of VHEE accelerators throughout the world (rows 350-447, more than 1 page), describing each separately may not be relevant to a journal article.
- Focusing of the beam is described in detail in 2,5 pages (rows 600 – 759). It is very detailed and may not be relevant to a journal article.
- Comparison VHEET and proton plans for prostate radiotherapy (rows 1109 – 1134). No advantage of VHEET is found, however the comparison is described in the extent of half page. It may be more useful to present simply no advantage has been achieved.
- Section 10 “Conclusions” up to row 1375 repeats again what has been mentioned in previous sections. However, there is a plausible listing from row 1379 on, which draws lucidly the main conclusions and future directions.
There is a plenty of abbreviations used in the text. A list of abbreviations explaining the sense of all may be useful.
The review presents 9 very illustrative figures which enable to understand the reader physics of VHEE well. Only the figure depicting the geography of VHEE sources throughout the world seem rather abundant in a scope of the journal article.
5 lucid tables give a comprehensive information. The tables could save much of the text that presents the information given in the tables. Only reference to the tables would be enough.
The list of 175 citations is relevant to the extent of the text. All citations are recently dated as relevant to the nature of the topic.
There are no ethic aspects of the review.
Conclusion
The presented white paper is too detailed, its comprehensiveness and extent compromise its lucidity. It is addressed to various specialties in “one package”, thus a lot of information may be found irrelevant.
The extent of the article is extremely long, not relevant to the periodic journal. All information given is well documented and presented in a logical order. The conclusions are very detailed however plausible.
The article is recommended to be reconsidered after major revision – reduction of the extent.
Author Response
Response to Reviewer 1
Thank you for your detailed feedback and for recognizing the comprehensiveness and logical structure of our review. We greatly appreciate your thoughtful evaluation and acknowledgment of the extensive body of literature covered.
Regarding the length and scope of the manuscript, we agree that it is extensive; however, this was a deliberate choice during its preparation. Our aim was to create a resource that caters to the diverse specializations within the VHEE research community, including experimental physicists, medical physicists, clinicians, and biologists. The comprehensive nature of the review ensures that specialists from various fields can find detailed and relevant information tailored to their specific needs.
We believe this multidisciplinary approach is crucial, as the development and clinical translation of VHEE technology require close collaboration among these fields. While not every section may be equally relevant to all readers, the review provides a foundation for specialists to understand how their work fits into the broader context of VHEE research.
To address concerns about lucidity, we have taken steps to improve the clarity and accessibility of the text, including restructuring certain sections for better readability (for instance, Section 5.1.1). These changes ensure that the information is easier to navigate while preserving the depth required for a thorough review.
We also acknowledge your comment regarding the title and have revised it by removing the term "study" to better reflect the manuscript's nature as a comprehensive review.
While we agree that including a list of abbreviations at the beginning could be beneficial and, in principle, have no objection to adding such a list, this is not stipulated by Cancers guidelines. The guidelines explicitly require that all acronyms be defined upon their first use in the abstract, main manuscript, and figures.
We hope these points address your concerns. Additionally, in a point-to-point response, we detail how we have rearranged and reduced the length of the indicated sections.
Comment 1: Description of flash effect and FLASH-RT (rows 262-285). It may be enough to refer to relevant publications. (A reader able to understand this survey definitely knows what means FLASH.)
Response 1: We agree with the reviewer. We made significant cuts in Section 4 (VHEE FLASH Radiotherapy). Specifically, we removed the lines covering 1) the history of FLASH studies, and 2) the mechanisms underlying the FLASH effect, as these topics are outside the scope of this paper. We also condensed the lines on the clinical advantages of FLASH-RT. However, we retained the section describing the temporal parameters required to trigger the FLASH effect, as this information is critical for understanding the technological aspects of VHEE FLASH-RT systems. Overall, this section has been significantly reduced.
Comment 2: The listing of VHEE accelerators throughout the world (rows 350-447, more than 1 page), describing each separately may not be relevant to a journal article.
Response 2: We agree with the reviewer and have significantly revised Section 5.1.1 (RF Linacs VHEE Beams Accelerators, Facilities Overview). The detailed descriptions of individual RF-based VHEE facilities have been removed, and this section now consists of a concise text summary accompanied by a table (Table 2) listing the relevant parameters for RF-based VHEE facilities where such data are available. This revision resulted in a reduction of nearly 2 pages. To facilitate a more targeted comparison, in the following section we have included Table 3, which presents the relevant parameters for laser-based VHEE facilities.
Additionally, in Section 5.2 (Laser-Driven VHEE Beams Accelerators), we removed lines more than 20 lines which discussed injection techniques in laser-plasma accelerators. These details were beyond the paper's scope, and we now just provide a reference to guide interested readers. We hope the reviewer finds these changes satisfactory.
Comment 3: Focusing of the beam is described in detail in 2,5 pages (rows 600 – 759). It is very detailed and may not be relevant to a journal article.
Response 3: We understand the reviewer’s concern. However, we believe that beam focusing is critical for significantly enhancing the dosimetric properties of VHEE beams. To address the comment, we condensed Section 6.1 (Focusing Monochromatic VHEE Beams from RF Linacs) into a more succinct text while ensuring no essential concepts were omitted. This revision reduced the section by 14 lines.
Comment 4: Comparison VHEET and proton plans for prostate radiotherapy (rows 1109 – 1134). No advantage of VHEET is found, however the comparison is described in the extent of half page. It may be more useful to present simply no advantage has been achieved.
Response 4: We understand the comment but respectfully disagree. While many studies compare VHEET with photon RT, only a few investigate VHEET versus proton RT. This section summarizes these limited studies to provide a complete picture of the available comparisons. We believe reducing this discussion to a single sentence would overlook its significance.
Comment 5: Section 10 “Conclusions” up to row 1375 repeats again what has been mentioned in previous sections. However, there is a plausible listing from row 1379 on, which draws lucidly the main conclusions and future directions.
Response 5: We agree that the Conclusions section should provide a concise summary of the key points outlined in the manuscript. As suggested, we reduced the extent of the first part of the Conclusions by over 10 lines while retaining the listing of conclusions and future directions from row 1379 onward.
We are grateful for your constructive feedback, which has helped us improve the manuscript. We hope the performed revisions address the reviewer’s concerns and result in a clearer, more concise manuscript. Thank you again for your valuable feedback.
On behalf of all authors,
Costanza Panaino
Reviewer 2 Report
Comments and Suggestions for Authors
The manuscript is a review of very high-energy electron therapy (VHEET). Considering that electron therapy currently appears to be at the forefront of potential implementation of FLASH therapy, this is an opportune moment for a comprehensive review of this treatment modality.
The manuscript is well-written, easy to follow, and extensive, covering a wide range of significant aspects of VHEET. It references a large number of key articles within the field, demonstrating substantial bibliographic effort and thorough research.
Author Response
On behalf of all authors,
We sincerely thank the reviewer for their positive and encouraging feedback on our manuscript. We greatly appreciate their recognition of the timeliness and significance of this review, as well as their kind words regarding its clarity, comprehensiveness, and extensive bibliographic effort.
Sincerely,
Costanza Panaino
Reviewer 3 Report
Comments and Suggestions for Authors
The authors give a comprehensive collection of the results of studies concerning VHEET and FLASH-VHEET. It contains many different fields of VHEET as accelerator technics, dosimetric beam properties, radiation protection, radio biology, dose measurement systems, treatment planning and comparison of different treatment technics. Therefore, it is impossible for a single person to check all the details of the cited literature in few days.
Nevertheless, the article can serve as an extensive overview of the different aspects of VHEET and can be used by the reader as the starting point for further studies of his/her interesting aspects.
Some minor comments:
· The authors use dmax (the depth of max. dose) also as the value of the maximal dose (e.g., line 79 ff: 90% of dmax). Better is to use “90% of the dose at dmax“ or “90 % of Dmax”.
· In line 259: In which time does one get this equivalent dose per Gy? What is the equivalent dose rate? What does it mean for the stuff? If it would be the value of 1 day for the stuff, it is a very large dose. It should be explained in more detail.
· At lines 829 and 835: TVA instead of TWA.
· In lines 904-906, the definition of RBE is misleading. The RBE of a radiation is the ratio of the dose of this radiation to the dose of a reference radiation that yield the same effect (not the same dose amount).
· In line 1281: NTID instead of NTDI.
· In fig. 6: What does “10s of fs” mean? Does it mean “about 10 to 90 fs”?
Author Response
We sincerely thank the reviewer for their kind and constructive comments on our manuscript. We greatly appreciate their recognition of our efforts to provide a comprehensive overview of the various aspects of VHEET and FLASH-VHEET research. Their thoughtful feedback and observations have been very helpful in refining our work and ensuring it serves as a valuable resource for the research community. A point-by-point response to the minor comments is provided below.
Comment 1: The authors use dmax (the depth of max. dose) also as the value of the maximal dose (e.g., line 79 ff: 90% of dmax). Better is to use “90% of the dose at dmax“ or “90 % of Dmax”.
Response 1: Thank you for pointing this out. We agree with this comment. We have defined d_max, as the depth of maximum dose and D_max as the maximum dose. We then checked for consistency in the use of these terms throughout the entire manuscript. Corrections were made in Sections 2.1, 2.2, 2.3, 3.1.1, 6.1, 6.2, and Figure 1.
Comment 2: In line 259: In which time does one get this equivalent dose per Gy? What is the equivalent dose rate? What does it mean for the stuff? If it would be the value of 1 day for the stuff, it is a very large dose. It should be explained in more detail.
Response 2: Thank you for pointing this out. We agree with this comment and have added a definition of the equivalent dose for clarification.
Comment 3: At lines 829 and 835: TVA instead of TWA.
Response 3: Thank you for pointing this out. We have corrected this typographical error in the manuscript.
Comment 4: In lines 904-906, the definition of RBE is misleading. The RBE of a radiation is the ratio of the dose of this radiation to the dose of a reference radiation that yield the same effect (not the same dose amount).
Response 4: Thank you for pointing this out. We agree with this comment and have revised the definition of RBE in the manuscript as suggested.
Comment 5: In line 1281: NTID instead of NTDI..
Response 5: Thank you for pointing this out. We have corrected this typographical error in the manuscript.
Comment 6: In fig. 6: What does “10s of fs” mean? Does it mean “about 10 to 90 fs”?
Response 6: Thank you for pointing this out. We understand that the phrase “10s of fs” may not be clear. We have revised it to read “~10–40 fs” for clarity. This parameter can also be seen in Table 3 (which was added following the advice of another reviewer)
We hope these revisions address the reviewer’s concerns and result in a clearer, more concise manuscript. Thank you again for your valuable feedback.
On behalf of all authors,
Costanza Panaino